# The CNS connectome of a tadpole larva of *Ciona intestinalis* (L.) highlights sidedness in the brain of a chordate sibling

Kerrianne Ryan[1,2], Zhiyuan Lu[1,2], Ian A Meinertzhagen[1,2]*

[1]Department of Biology, Life Sciences Centre, Dalhousie University, Halifax, Canada; [2]Department of Psychology and Neuroscience, Life Sciences Centre, Dalhousie University, Halifax, Canada

**Abstract** Left-right asymmetries in brains are usually minor or cryptic. We report brain asymmetries in the tiny, dorsal tubular nervous system of the ascidian tadpole larva, *Ciona intestinalis*. Chordate in body plan and development, the larva provides an outstanding example of brain asymmetry. Although early neural development is well studied, detailed cellular organization of the swimming larva's CNS remains unreported. Using serial-section EM we document the synaptic connectome of the larva's 177 CNS neurons. These formed 6618 synapses including 1772 neuromuscular junctions, augmented by 1206 gap junctions. Neurons are unipolar with at most a single dendrite, and few synapses. Some synapses are unpolarised, others form reciprocal or serial motifs; 922 were polyadic. Axo-axonal synapses predominate. Most neurons have ciliary organelles, and many features lack structural specialization. Despite equal cell numbers on both sides, neuron identities and pathways differ left/right. Brain vesicle asymmetries include a right ocellus and left coronet cells.

*For correspondence: iam@dal.ca

**Competing interests:** The authors declare that no competing interests exist.

## Introduction

Animals exhibit various forms of left-right asymmetry (*Ludwig, 1932*; *Neville, 1976*; *Brown and Wolpert, 1990*; *Palmer, 2009*), and anatomical examples in brains are reported both in invertebrates (*Hobert et al., 2002*; *Frasnelli et al., 2012*), and vertebrates (*Rogers and Andrew, 2002*; *Duboc et al., 2015*) as differences in cell number (*Blinkov and Glezer, 1968*, their tables 184 and 185) and particularly in the lateralization of the mammalian cortex (*Galaburda et al., 1978*). Even predominantly symmetrical bilaterians exhibit various forms of sidedness, in both protostomes (*Grande and Patel, 2009*) and deuterostomes (*Hamada et al., 2002*; *Duboc et al., 2005*). In large, complex brains, these anatomical asymmetries are often structurally cryptic, and gene expression has been used as a proxy to examine the evolution of sidedness. As a sister group to vertebrates (*Cameron et al., 2000*; *Satoh et al., 2014*), ascidians have larvae that are chordate in body plan and mode of development, with a dorsal central nervous system (CNS). In the ~1 mm tadpole larva of *Ciona intestinalis* neurons are distributed rostrocaudally in three main centres, a brain vesicle, motor ganglion and caudal nerve cord (*Katz, 1983*; *Nicol and Meinertzhagen, 1991*; *Meinertzhagen et al., 2004*). The CNS forms from a neural tube, yet exhibits left/right differences, and so provides a useful model to study many aspects of brain asymmetry. This issue is important because brain laterality has been associated with increased fitness for animal life (*Duboc et al., 2015*).

**eLife digest** Brains are made up of a network of nerve cells (neurons) that are connected to each other by junctions called synapses. The neurons on the left and right sides of the brain form different patterns of connections, but this asymmetry can be difficult to spot because the brain is large and complex. Understanding how the whole network operates is key to understanding how the brain works. However, a full map of all the connections between neurons – known as a connectome – has only been described for one species so far, a nematode worm called *C. elegans*.

The tadpole larva of the common sea squirt has a fairly simple brain distantly related to our own but made up of only about 330 cells. Ryan et al. used a technique called electron microscopy to study thin sections from the brains of sea squirt larvae to reveal this animal's connectome and investigate left-right asymmetry in the brain.

The analysis revealed 177 neurons in this larval brain, just over half of its brain cells. These can be split into at least 25 types and each neuron has a simple, mostly unbranched shape with, on average, 49 synapses with other cells. This means that, even though it has such a small number of neurons, the neuron network is still relatively complex. The shortest sensory pathway to any muscle connects via three synapses, although most pathways involve more. The left and right sides of the brain differ in the types of neurons they contain and the connections these form, even though both sides have the same number of cells.

The findings of Ryan et al. reveal the second animal connectome and lay the groundwork for future studies on how each neuron in the network influences the behaviour of the sea squirt's larva. Further work is also required to find out how the patterns of synapses in the brain change as the larva ages, and whether the connectome differs between siblings.

The most studied tunicate species is *Ciona intestinalis* (*Satoh, 1994*). Not only does its development result from a fixed pattern of cell lineage and result in a mere ~ 2600 cells in the larva of *Ciona intestinalis* (*Satoh, 1999*), but the genome, first in *Ciona intestinalis* (*Dehal et al., 2002*) and now in nine other species (*Brozovic et al., 2016*), has been sequenced. Even though the events of early neural development and the nervous system's subsequent metamorphosis have been identified, together with many of their underlying causal gene networks (*Satoh, 2003*; *Sasakura et al., 2012*), the detailed cellular organization of their product, the CNS of the swimming larva, still remains almost entirely unresolved.

*Ciona* releases 5000–10000 eggs per individual (*Petersen and Svane, 1995*), and its eggs are released either individually, or in a mucous string (*Svane and Havenhand, 1993*). Gametes undergo fertilization, cleavage, development, and then hatch into non-feeding lecithotrophic larvae in the water column. Initially after hatching, larvae swim up toward the surface of the water by negative geotaxis using the otolith cell (*Tsuda et al., 2003*) a behaviour retained in ocellus-ablated larvae. Later in larval life, larvae exhibit negative phototaxis, swimming down to find appropriate substrates for settlement (*Tsuda et al., 2003*). The swimming period exhibits three characterized behaviours: tail flicks (~10 Hz), 'spontaneous' swimming (~33 Hz), and shadow response (~32 Hz; *Zega et al., 2006*). Larvae swim more frequently and for longer periods earlier in life up to 2 hr post hatching (hph). Of the reported behaviors, the shadow response, in which a dimming of light results in symmetrical swimming, is the best studied, developing at 1.5 hph and increasing in tailbeat frequency after 2 hph (*Zega et al., 2006*). In addition to phototactic and geotactic behavior, there is evidence of chemotactic behavior just before settlement (*Svane and Young, 1989*) and of some mechanosensory responses in swimming larvae (*Bone, 1992*). Because larvae do not feed, their main biological imperative is survival and successful settlement to undergo metamorphosis into a sessile adult, in an environment with appropriate food and reproductive resources. Thus, entering the water current and avoiding predation by filter feeders may be the foundation for the larva's many behavioral networks, especially in early life before settlement.

The substrate for these behaviours is the larva's dorsal central nervous system, which is divided into the anterior sensory brain vesicle (BV), connected by a narrow neck to the motor ganglion (MG) within the larval trunk, and a caudal nerve cord (CNC) in the tail (*Nicol and Meinertzhagen, 1991*).

Sensory neurons of the CNS and their interneurons reside in the BV, which has an expanded neural canal and the most complex neuropil. The relay neurons of the posterior brain vesicle extend axons through the neck to the motor ganglion, which overlies the anterior portion of the notochord, and contains neurons of the motor system. At the trunk-tail border, muscle cells of the tail flank the notochord and CNS, and these extend down through the tail alongside the narrow, simple CNC. In addition to the CNS several sensory epidermal neurons (ENs) of the peripheral nervous system (PNS) populate the dorsal and ventral axes of the larva in a rostrocaudal sequence, with axons running beneath the epidermis (*Imai and Meinertzhagen, 2007b*).

Many asymmetries have been uncovered by the developmental expression of Nodal and its signaling pathways (*Hamada et al., 2002*; *Hudson, 2016*). As in vertebrates, in ascidians, their sibling group (*Satoh et al., 2014*), Nodal expresses on the left hand side of the developing embryo (*Boorman and Shimeld, 2002a, 2002b*; *Yoshida and Saiga, 2008*). This is true neither of other deuterostomes (*Duboc et al., 2005*) nor lophotrochozoans (*Grande and Patel, 2009*), while ecdysozoans such as *Drosophila* and *C. elegans* lack Nodal (*Schier, 2009*), even though the brain in *Drosophila* is asymmetrical (*Pascual et al., 2004*). The development of brain asymmetry in the ascidian does however depend on the presence of an intact chorion in the embryo (*Shimeld and Levin, 2006*; *Yoshida and Saiga, 2008*; *Oonuma et al., 2016*).

In contrast to the situation in most chordates, structural brain asymmetries, which include cell numbers, positions, and connections are externally visible in the tadpole larva of ascidians, for example from the pigment spots and right-sided ocellus in the head of *Ciona intestinalis* (*Eakin and Kuda, 1971*; *Katz, 1983*; *Nicol and Meinertzhagen, 1991*). Photoreceptor neurons associated with the ocellus are of the ciliary type, with outer segment lamellae orientated parallel to the cilium (*Eakin and Kuda, 1971*), in contrast to the perpendicular arrangement found in vertebrate rods (*Lamb and Collin, 2007*). The photoreceptor cells of the ocellus on the right of the brain vesicle, which number 20 cells (*Horie et al., 2008*), 17 and 18 of which had been reported by *Nicol and Meinertzhagen (1991)*, are twinned with 17 to 19 structurally different coronet cells (previously claimed hydrostatic pressure receptor cells: *Eakin and Kuda, 1971*; *Nicol and Meinertzhagen, 1991*) on the left. This sidedness may also correspond to larval behavior because ascidian larvae pursue a helical trajectory when swimming (*McHenry and Strother, 2003*; *McHenry, 2005*) and ascidian larvae are thought to use klinotaxis to respond to visual cues by modulating the symmetry of tail kinematics (*McHenry and Strother, 2003*). The pattern of helical swimming arises from bilateral contractions of the tiered muscle bands on either side of the notochord. On each side, gap junctions connect all 18 uninucleate muscle cells, arranged in three rows, dorsal medial and ventral (*Bone, 1992*). Muscle activity along the tail is thus not segmental, and instead propagates posteriorly in a wave from innervated anterior dorsal and medial muscle cells at the trunk-tail border without the requirement for additional neuronal input along the tail (*Bone, 1992*).

In the past, the search for brain asymmetries has been frustrated by two features, the presumed relative rarity of such asymmetries, and the lack of structurally identified networks of neurons in which to recognise them. These obstacles are resolved in the CNS of the tadpole larva of *Ciona*, in which not only is the asymmetrical organization obvious, but the possibility exists to uncover the larva's complete network of neuronal synaptic connections, or its connectome (*Lichtman and Sanes, 2008*). That possibility rests on the tiny size of the brain, the small number of its constituent cells, approximately 330 (*Nicol and Meinertzhagen, 1991*), and the morphological simplicity of its neurons (*Imai and Meinertzhagen, 2007a*). The apparent symmetry of the characteristic swimming pattern of chordate-like tail undulations (*Bone, 1992*; *Video 1*) stands in marked contrast to the asymmetries in the CNS that generates them. Here we present the connectome of the CNS of the tadpole larva in *Ciona*, and reveal its asymmetries in the left and

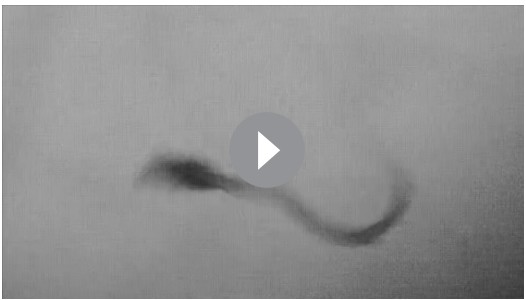

**Video 1.** Symmetrical undulations of the tail in a swimming *Ciona* larva. The tail lacks segmentation and in the 2 hr hatchling larva oscillates at 20–30 Hz at the juncture with the rostral trunk (*Bone, 1992*).

right complements of neurons and their synaptic networks.

## Results

A single larva was prepared for electron microscopy (EM) and a long continuous series of cross sections cut. A total of 3375 60 nm sections that started at the level of the otolith pigment and extended to the posterior motor ganglion, followed by 1360 70 nm sections cut into the anterior tail, and continued to its tip by a further 2193 100 nm sections (*Figure 1*; *Figure 1—figure supplement 1*) were collected and imaged at 3.85 nm per pixel. The entire imaged series thus included the anterior brain vesicle and motor ganglion (MG) cut at 60 nm, and the caudal nerve cord at 70, then 100 nm. These regions were additionally innervated by neurons of the PNS with cell bodies residing in the epidermis; these extend cilia into the tunic. Thus, unlike *C. elegans*, which lacks a distinction between a CNS and PNS, *Ciona*'s chordate nervous system has clear divisions between the two. We annotated mostly those PNS neurons that innervate the CNS (*Figure 1A*).

### Synapses

Synapses were identified based on criteria established in other invertebrate species (*White et al., 1986*; *Westfall, 1996*; *Meinertzhagen, 2016*): primarily a cluster of vesicles at a presynaptic membrane. Small electron-lucent vesicles 30 to 60 nm in diameter were found exclusively at some presynaptic sites where they tended to cluster closely to form a tight cumulus (*Figure 2A*), accompanied at some synapses by larger electron-lucent vesicles (70—110 nm) (*Figure 2B*). The same vesicle types were found throughout the neuron, but their distribution was not quantified. Some synapses also had dense-core vesicles, large (110–140 nm; *Figure 2C*), medium (70–80 nm; *Figure 2B*), or small (40–60 nm) (*Figure 2D*; *Figure 2—source data 1*). The size of the internal density within these vesicles varied, some medium sized dense-core vesicles having small cores (*Figure 2E*). Other synapses had exclusively medium to large dense-core vesicles. Postsynaptic densities (*Gray, 1959*; *Peters and Palay, 1996*) were observed at some synapses (*Figure 2C*) but not at all, thus providing an unreliable criterion for a synaptic contact.

Membrane appositions interpreted as putative gap junctions (*Bennett and Goodenough, 1978*) were annotated for contacts at which densities were present on the membranes of both sides (*Figure 2F*), except where such contacts formed directly adjacent to the neural canal, thus excluding junctions provisionally interpreted as desmosomes or adherens junctions.

Synapses and gap junctions varied in size, contributing profiles to between 1 and 29 sections (<66 sections for neuromuscular junctions) or 55 sections (gap junctions)(*Figure 3A*).

A total of 301 cells of the CNS were imaged, of which the CNS included 177 neurons with axons and presynaptic sites (*Figure 1—source data 1*; *Figure 3—source data 1*), and constituting the remainder, ependymal cells (those ciliated cells abutting the canal that lack an axon) and two cells of the CNS that are ambiguous, having presynaptic sites, but lacking a neuronal form (*Figure 1—source data 1* and *Figure 3—source data 1*). Cells omitted from our EM series, those rostral to the otolith and caudal to the bipolar tail neurons (*Imai and Meinertzhagen, 2007b*; *Stolfi et al., 2015*), are presumed to account for the remainder of the >331 cells reported by *Nicol and Meinertzhagen (1991)* and thus to number at least 30. This assumes the constancy of cell number between different sibling batches of larvae. Between the CNS neurons (and the four bipolar tail neurons), we identified 8768 synapses (6618 >1 section), including 1772 neuromuscular synapses, and 2105 putative gap junctions (1206 >1 section). Each CNS neuron thus formed on average 49 (standard deviation, SD 61) presynaptic sites with a range of between 1 and 430 synapses and an average of 13 (SD 23) putative gap junctions, with a range between 0 and 166. Each postsynaptic neuron received an average of 39 (SD 42) synapses in total from all its presynaptic partners, with a range between 0 and 179.

For each neuron, the number of presynaptic sites varies with the number of its postsynaptic partners, plotted for all neurons and their synapses but excluding the neuromuscular junctions (*Figure 3C*). This relationship indicates that with each additional partner, the number of synapses made by a presynaptic neuron increases, thus reflecting a postsynaptic drive to the total synapse load. There was no overall relationship for each neuron between the volume of its soma and the number of its synapses ($r^2 = 0.4$), nor between soma volume and axon/terminal surface area ($r^2 = 0.4$; *Figure 3—figure supplement 1*). However, axon/terminal surface area and number of synapses were weakly correlated ($r^2 = 0.7$) (*Figure 3—figure supplement 1C*).

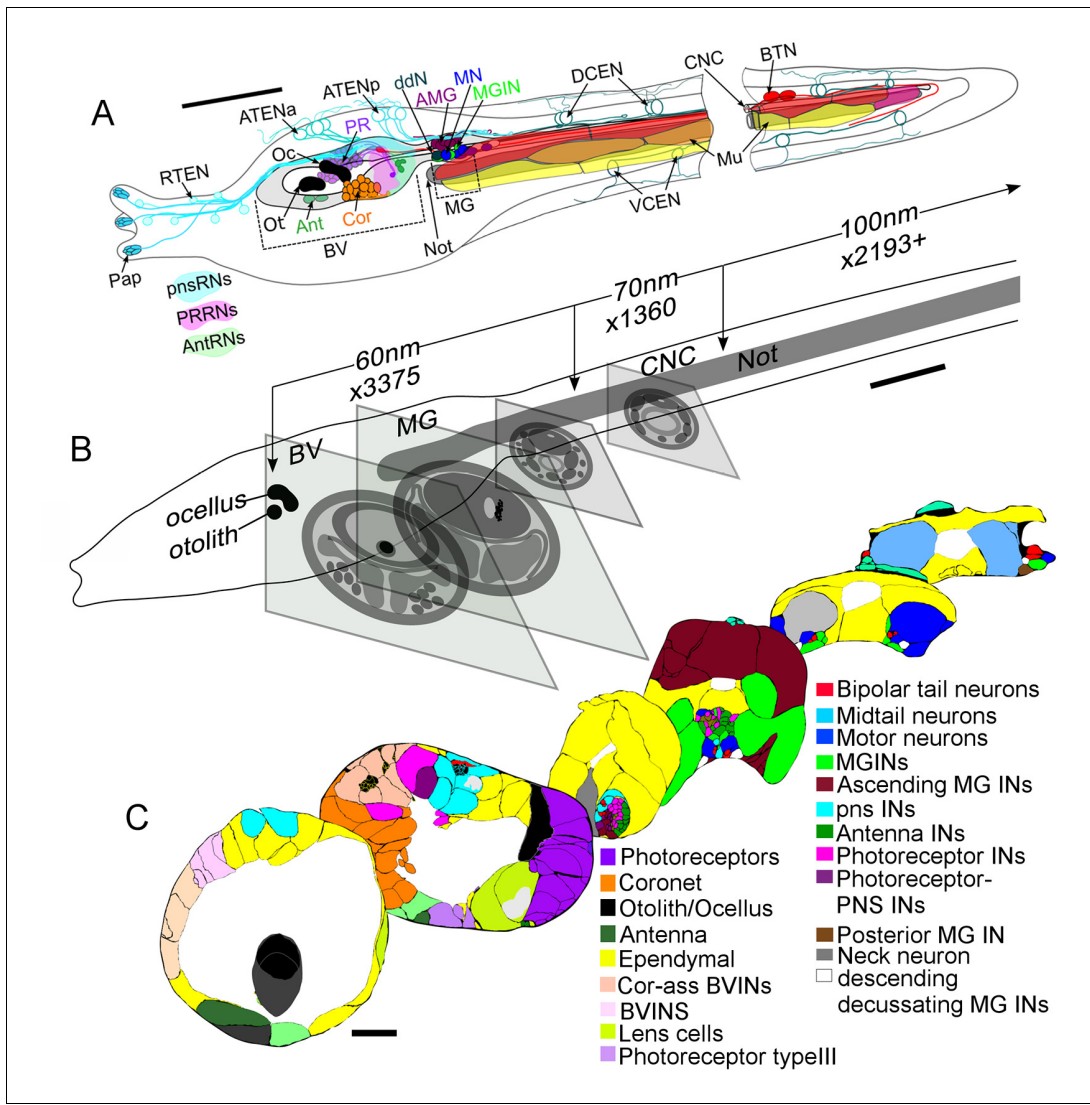

**Figure 1.** Ultrathin section series of *Ciona intestinalis* larva, its CNS and notochord. (**A**) Diagram of whole larva with colour-coded cell types indicated by arrows. Types of relay neurons (RNs) are shown as colour-coded territories in the brain vesicle. Muscle cells align in dorsal (red), medial (orange), and ventral (yellow) tiers. (**B**) Larva from left side, illustrating major landmarks and indicating the number of sections of each thickness (60–100 nm) along the A-P axis. Diagrammatic profiles of representative sections at different levels along the A-P axis of the larva, shown to the same magnification. (**C**) Profile traces from *Reconstruct* of representative sections at six levels of the CNS. Nuclei are shown as small squares. Coloured cell types are shown in the key (*Figure 1—source data 1*). For labelled cell outlines of the sections see enlarged views in *Figure 1—figure supplement 1*. Abbreviations: Pap: papilla neuron: RTEN: rostral trunk epidermal neurons; ATENa: anterior apical trunk epidermal neurons; ATENp: posterior apical trunk epidermal neurons; Oc: ocellus; Ot: otolith; Ant: antenna neuron; Cor: coronet cell; PR: photoreceptor; ddN: descending decussating neuron; AMG/Ascending MG IN: ascending motor ganglion interneuron: MGIN: motor ganglion interneuron; MN: motor neuron; DCEN: dorsal caudal epidermal neuron; VCEN: ventral caudal epidermal neuron; BTN: bipolar tail neuron; Mu: muscle; pnsRNs: PNS relay neurons; PRRNs: photoreceptor relay neurons; AntRNs: antenna relay neurons; BV: brain vesicle; MG: motor ganglion; CNC: caudal nerve cord; Not: notochord; IN: interneuron; BVINs: brain vesicle interneurons. For descriptions and abbreviations of cell types also see key in *Figure 1—source data 1* Scale bars in all panels: 10 μm.

The following source data and figure supplement are available for figure 1:

**Source data 1.** Cell types: key to their characteristics and abbreviations used.

**Figure supplement 1.** Enlarged representative sections with labeled profiles from *Figure 1B*.

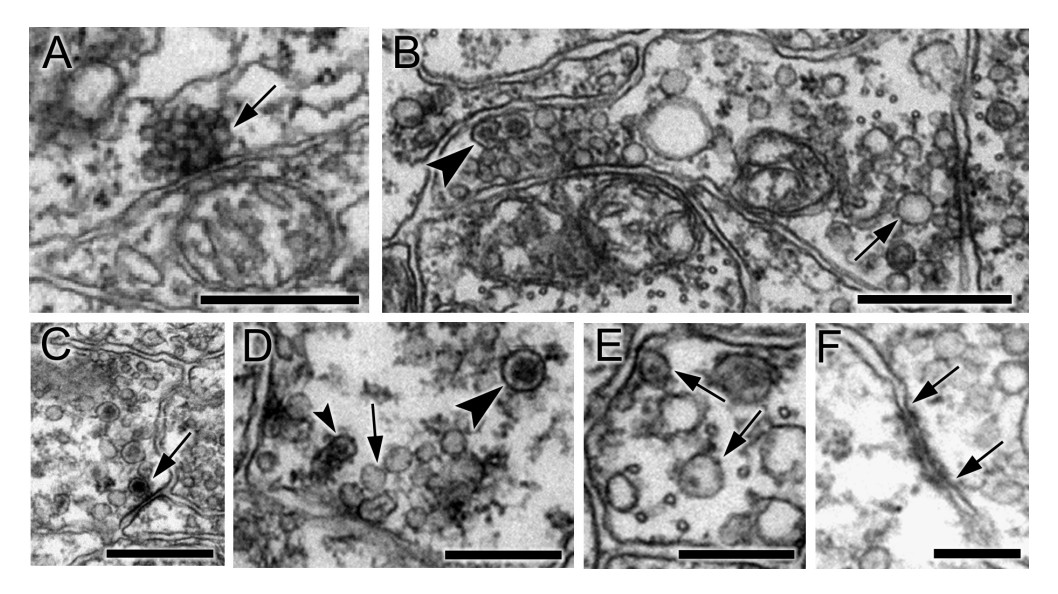

**Figure 2.** Synapses contain presynaptic vesicles of various sizes and types. (**A**) Tightly packed cumulus of small (30–40 nm) vesicles at a single presynaptic site (arrow). (**B**) Mixed populations of small (30–50 nm) and large (70–110 nm) electron-lucent vesicles (arrow) as well as dense-core vesicles of medium size (arrowhead). (**C**) Large (100–110 nm) vesicles with dark cores (arrowheads). (**D**) Synapses containing electron-lucent vesicles (30–60 nm; arrow) as well as small (60 nm; small arrowhead) and medium (80 nm; large arrowhead) dense-core vesicles. See *Figure 2—source data 1* for a list of those neurons with synapses having mixed vesicle populations. (**E**) Medium dense-core vesicles with small cores (arrows). (**F**) Membrane apposition (between arrows) interpreted as a gap junction, with membrane densities on both sides. Scale bars: 500 nm (A,B,C and E); 200 nm (D and F).

The following source data is available for figure 2:

**Source data 1.** Neurons with synapses having mixed electron-lucent and dense-core vesicle populations (mixed), or exclusively dense-core vesicle (dcv) populations, with numbers (No.) of synapses of each type, and totals of both dcv and mixed vesicle synapses.

Synaptic structure is sometimes unpolarised (*Figure 4A*), with synaptic vesicles situated on either side of the synaptic cleft; neurons also frequently connected to form both reciprocal (*Figure 4B*) and serial synapses (*Figure 4C*; *Table 1*). Of the 8617 synapses, 922 were polyadic, having multiple post-synaptic elements (*Figure 4C*; *Figure 4—source data 1*; *Table 1*). The most common of these were dyads, which constituted 93% of all polyadic synapses, and were common especially in antenna neurons (see below).

The network forms a single connected component, with all cells (nodes) being connected by a synapse (edge) to another node in the network (*Figure 3—figure supplement 2*). The network statistics (*Table 2*) reveal that the characteristic path length between two neurons is 2.7 (from one neuron, through one other to its target), with neurons having an average of 20 neighbors (synaptic partners) and an overall average network clustering coefficient (existing edges between neighbors of a neuron/possible edges between neighbors of a neuron) of 0.333.

## Neurons

Within the CNS the distribution of presynaptic sites over the surface of the neuron varies by cell type. Most neurons are monopolar, <25% only having dendrites, and their axons usually form a clear terminal. Axons fasciculate in bundles but braid their positions within their bundle or sometimes defasciculate. Brain vesicle (BV) intrinsic interneurons have approximately equal numbers of presynaptic sites over their axons as their terminals, each constituting approximately 40% of their total. In contrast, axo-axonal synapses are the predominant synapses involving relay and motor ganglion (MG) interneurons, comprising >50% for both ascending and descending MG interneurons

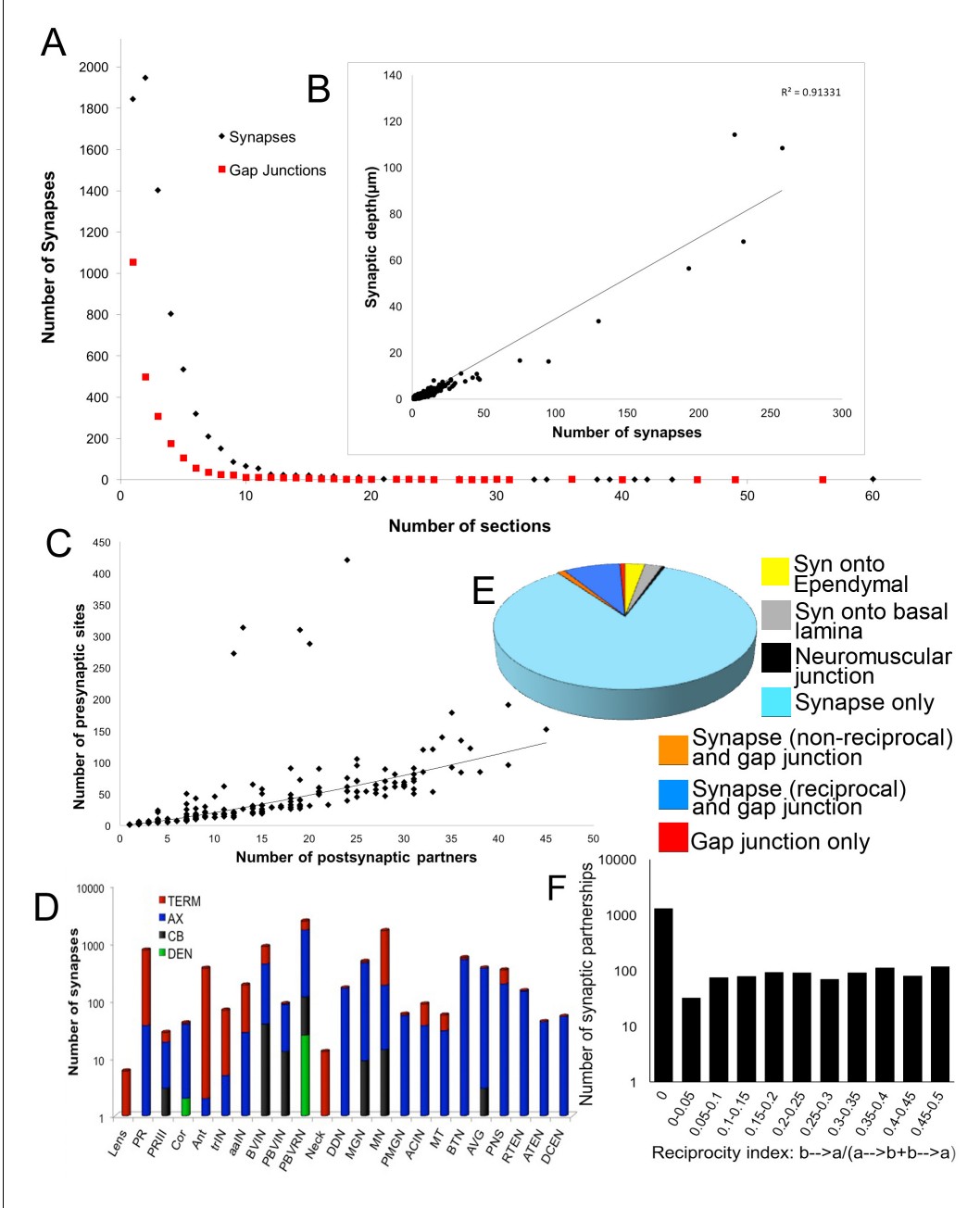

**Figure 3.** Synapse numbers (presynaptic sites) and sizes for all neurons (for complete list see *Figure 3—source data 1*). (**A**) Most synaptic contacts extend over <10 60 nm sections. Those occupying >10 sections are neuromuscular junctions, inputs from relay neurons to MG neurons, and synapses from antenna cells. The frequency curve for chemical synapses reveals more large contacts than for gap junctions. (**B**) Plotted for all neurons, the total depth of presynaptic contact co-varies linearly with the total number of synapses ($R^2$ = 0.91). Removing single-profile synapses eliminates 18% of all synaptic partnerships, and removing all 2-profile synapses would have eliminated a total of 35% of all synaptic partnerships. (**C**) The number of presynaptic sites co-varies with the number of postsynaptic partners according to a power function ($R^2$ = 0.81). The number of synaptic partners is also referred to as the network statistic 'degree', and is mapped to the synaptic network (*Figure 3— figure supplement 2*). Five neurons lie well above the curve, having low degrees, with many synapses and few postsynaptic partners. These are: Antenna neuron 1 (Ant1) with many synapses onto seven relay neurons, and the two pairs of anterior-most motor neurons (MN1 and MN2) with many synapses onto muscle. (**D**) Cumulative distribution of the log number of presynaptic sites over the surfaces of neurons of major cell classes. TERM: terminal; AX: axon; CB: cell body; DEN: dendrite. Most synapses are located over axons and terminals (see

*Figure 3 continued on next page*

*Figure 3 continued*

**Figure 3—figure supplement 1** for (D) averages per neuron and (E) postsynaptic site distribution). (E) Proportions of synapses and gap junctions in the connectome formed for particular partnerships. (F) Reciprocity of connections in the network given as the proportion of neuron partners that are reciprocally connected and the extent of their reciprocity (calculated as the cumulative depth of contacts in one direction divided by the sum of the depth of all contacts between the neuron pair). The total proportion of reciprocal synaptic connections between neuron pairs is 0.39.

The following source data and figure supplements are available for figure 3:

**Source data 1.** Summary of all neurons in the larval CNS of *Ciona intestinalis.*
**Figure supplement 1.** Relationships between the morphology and synaptic output of larval CNS neurons.
**Figure supplement 2.** Network graphs with network statistics visualized as attributes.

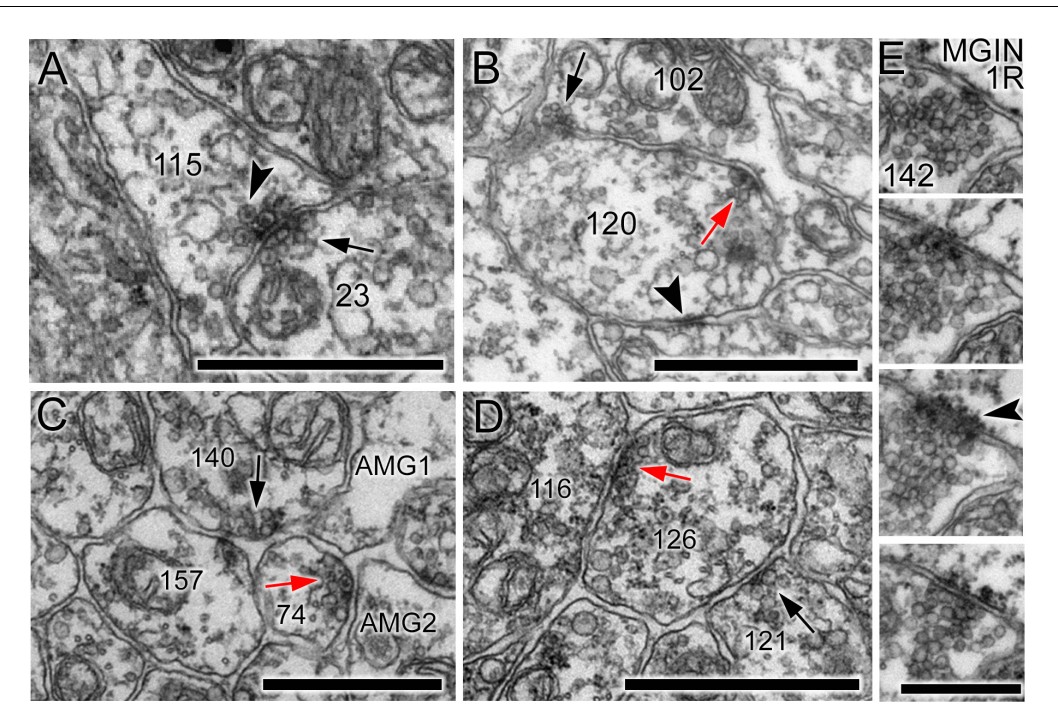

**Figure 4.** Unpolarized, reciprocal, and serial synapses. (A) Unpolarized mixed synapse between cell 115 and cell 23 with dense-core (arrowhead) and electron-lucent (arrow) vesicles on both sides of the synaptic cleft. (B) Single section with synapse from cell 102 to cell 120 (black arrow) and a reciprocal partner synapse from cell 120 to cell 102 (red arrow). Arrowhead: membrane apposition marking a putative gap junction. (C) Serial dyad synapse (black arrow) from single neuron onto two postsynaptic targets (157 and 74), one of which is presynaptic in the same section (red arrow) at a dyad synapse onto two neurites (AMG1 and AMG2). (D) Serial monad synapse (black arrow) onto a single postsynaptic target (126) that is presynaptic at an adjacent synapse (red arrow) to a single postsynaptic target (116). Scale bars: 1 μm. (E) Series of four 60 nm sections through a single synapse. The pre- and postsynaptic cell are labelled in the top image. A clear cumulus of presynaptic vesicles is visible all images, and a clear postsynaptic density in the penultimate image (arrowhead). Scale bar: 500 nm.

The following source data is available for figure 4:

**Source data 1.** Comparison of synaptic complements using different parameters and exclusion criteria.

**Table 1.** Numbers of synapses and gap junctions.

| | Total no. | Total no. sections | Mean no. sections /contact | No. synapses> 1 section | Mean no. sections /contact >1 section | % Unpolarized | % Polyad | % Dcv |
|---|---|---|---|---|---|---|---|---|
| Synapses | 8617 | 30163 | 3.5 | 6618 | 4.3 | 5.2 | 10.7 | 8 |
| Gap junctions | 3205 | 5765 | 1.8 | 1206 | 3.1 | ? | 3 | N/A |

Percentage (%) refers to the percentage of all synapses that are unpolarized (presynaptic vesicles on either side of the cleft between both neuron partners); polyadic (having >1 postsynaptic neurite); or containing dense-core vesicles (dcv) at the presynaptic site.

and >35% for BV relay neurons. The abundance of *en passant* synaptic contacts in the larval CNS of *Ciona* is particularly apparent when examining the proportional distribution of postsynaptic sites. Synapses formed by BV relay neurons at their terminals form *en passant* onto the axons and terminals of both relay and MG neuron targets, which together constitute >20% of all BV relay neurons' presynaptic contacts. Likewise, >50% of BV intrinsic interneuron synapses form *en passant* from their axons or terminals onto the axons or terminals of their targets. Synapses from terminal to terminal also constitute the greatest proportion of photoreceptor synapses (43%), and *en passant* synapses constitute over 60% of their contacts.

Overall, within the CNS and among the axons of the dorsal PNS, 68% of all neuron-neuron synapses terminate on axons or terminals. More synaptic contacts are made upon axons than upon terminals, the latter constituting only 23%–26% of all synapses, whereas those onto axons comprise 42%–44%. These numbers support the impression that presynaptic sites are formed at various places over the cell surface, and that each cell type has a preferred location to form them, but that this location is neither absolute nor exclusive. For further information see *Figure 3* and *Figure 3—figure supplement 1D*

**Table 2.** Network statistics for networks of chemical synapses and putative gap junctions.

| Statistic | Synaptic network | | Gap junction network | |
|---|---|---|---|---|
| | Full network | CNS neurons only | Full network | CNS neurons only (>0.06 μm) |
| Clustering co-efficient | 0.333 | 0.335 | 0.25 | 0.305 |
| Connected component | 1 | 1 | 7 | 1 |
| Network diameter | 9 | 7 | 8 | 8 |
| Radius | 1 | 4 | 1 | 4 |
| Shortest paths | 90% [41001] | 95% [29759] | 85% [31536] | 100% [16770] |
| Characteristic path length | 2.684 | 2.541 | 3.078 | 2.775 |
| Average number of neighbours | 20.169 | 20.689 | 8.674 | 10.369 |
| Number of nodes | 213 | 177 | 193 | 130 |
| Network density | 0 | 0 | 0.045 | 0.08 |
| Network heterogeneity | - | - | 0.935 | 0.76 |
| Number of self-loops | 19 | 16 | 13 | 9 |
| Multi-edge node pairs | 826 | 699 | 30 | 22 |
| Network centralisation | - | - | 0.191 | 0.257 |

Network statistics calculated using the Cytoscape Network Analyzer for network of chemical synapses (Synaptic network) and putative gap junctions (Gap Junction network) for both the full network thus including PNS neurons, muscle, ambiguous cells, and synapses onto basal lamina, as well as CNS neurons; and the network for CNS neurons only (CNS neurons). Note that the 'CNS neurons only' network excludes one additional isolated profile of a single branch of one photoreceptor terminal, probably pr10.

**Table 3.** Numbers of cells in the left, right and centre of the CNS and PNS.

| | Left | Centre | Right |
|---|---|---|---|
| Lens cells | | | 3 |
| Pigment cells | | | 2 |
| Total: Pigment and lens cells | | | 5 |
| Coronet | 13* | 2 | 1 |
| Photoreceptors | | | 37* |
| Antenna neurons | | 1 | 1 |
| Photoreceptor tract interneurons | | | 3* |
| Anterior BV neurons | 29 | 1 | |
| BV peripheral interneurons | 4 | 4 | 1 |
| Bipolar neurons | | | 2 |
| Anaxonal arborizing neurons | | 1 | 2 |
| Posterior BV peripheral interneurons | 2 | 1 | 1 |
| Photoreceptor relay neurons | 6 | | |
| Photoreceptor-peripheral relay neurons | 2 | 2 | 6 |
| Photoreceptor-coronet relay neurons | 2 | 1 | |
| Antenna-coronet relay neuron | | | 1 |
| Antenna relay neurons | 7 | | 2 |
| Peripheral relay neurons | 2 | 1 | |
| Relay interneurons | 5 | | |
| Total: BV neurons | 72 | 14 | 57 |
| Neck neurons | 1 | | 1 |
| Total: Neck neurons | 1 | | 1 |
| Ascending MG peripheral interneurons | 3 | 1 | 3 |
| Descending decussating neurons | 1 | | 1 |
| MG interneurons | 3 | | 3 |
| Motor neurons | 5 | | 5 |
| Total: MG neurons | 12 | 1 | 12 |
| Ascending contralateral inhibitory neurons (ACINs) | 2 | | 1 |
| Posterior MG interneurons | | | 2 |
| Mid-tail neurons** | 2 | | 2 |
| Total: CNC neurons | 4 | | 5 |
| All CNS neurons | 88 | 15 | 75 |
| Peripheral nervous system | | | |
| Bipolar tail neurons | 2 | | 2 |
| Peripheral neurons (RTENa) | 6 | | 6 |
| anterior ATENs | | 2 | 2 |
| posterior ATENs | | | 4 |
| DCENs | | 4 | |
| Total: PNS neurons | 8 | 6 | 14 |

Neurons of the left side of the nervous system outnumber those of the right, which in turn outnumber those of the centre. All CNS neurons include known neurons that lack synapses (*).

**Additional mid-tail neurons which lay beyond the analysed region of the EM series are excluded from the totals.

## Asymmetry in cellular composition

The overall cell complement including neurons, ependymal and accessory cells, is closely similar on the two sides (left: 125; right: 129; midline 46). The brain vesicle has unequal numbers of neurons and pigment cells, however, with more sensory neurons and pigment cells on the right side and more interneurons on the left (*Table 3*). Many of these left-side interneurons were previously identified as ciliated ependymal cells (*Nicol and Meinertzhagen, 1991*), but have been seen here to possess axons and synapses that were not visible by light microscopy, and are thus to be considered neurons. Excluding ependymal cells, each brain region has approximately equal numbers of cells on both sides (*Table 3*). Despite this near equal distribution in their overall numbers, however, examples of asymmetries in the numbers of identified neuron types, or numbers of neurons of each type, were nevertheless also found amongst specific interneuron classes in the brain vesicle. In the ventral motor ganglion most neurons are paired, including five pairs of motor neurons and four pairs of interneurons, whereas caudally among the ascending contralateral interneurons, ACINs, there were two representatives on the left and one on the right, as well as two descending posterior motor ganglion neurons (PMGNs) found only on the right (*Table 3*).

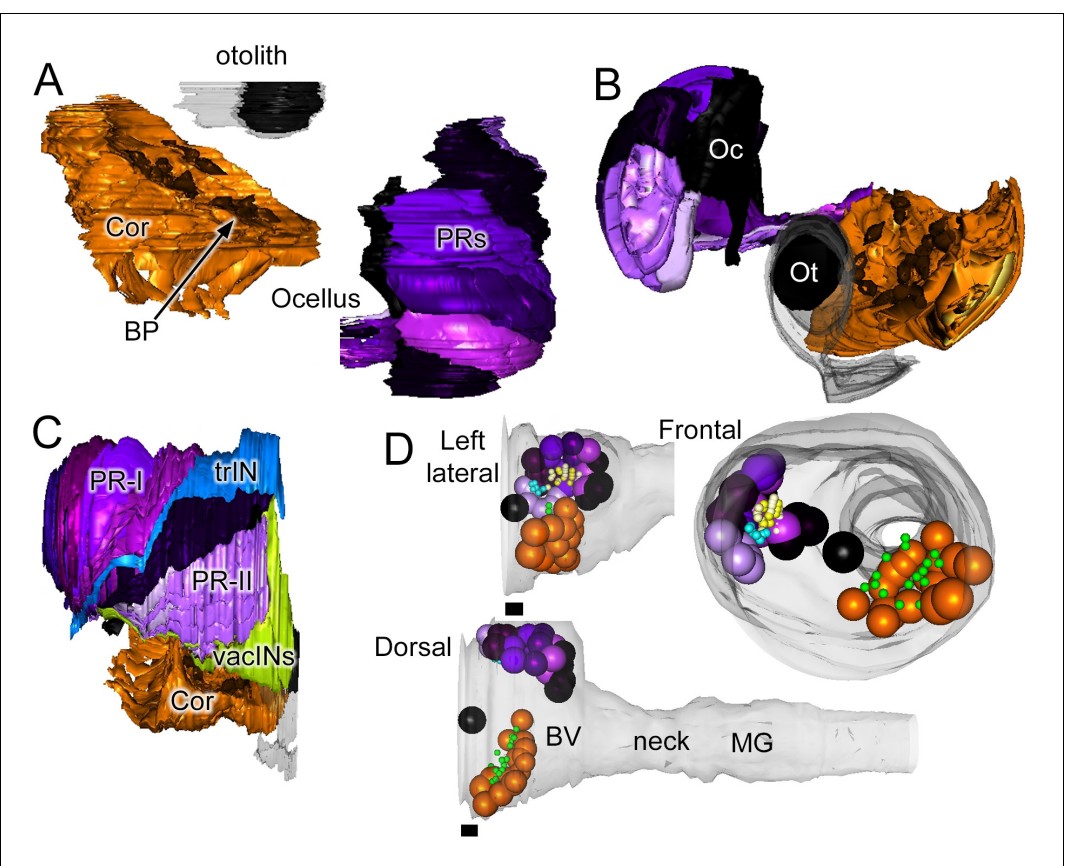

**Figure 5.** Sensory neurons and associated cells have sided distributions. Reconstructed coronet cells (Cor) with their bulbous protrusions (BP, one with a black arrow) and – in their correct relative position – six layers of photoreceptor neurons, excluding their terminals, together with otolith (Ot) and ocellar (Oc) pigment cells. Reconstructions shown from a dorsal (**A**), or frontal (**B**) view. (**C**) Reconstruction of visual system components (including photoreceptor tract (trIN) and vacuolated sensory (vacIN) interneurons, shown from the right side, anterior to the right. (**D**) Sensory neurons (spheroids) and their modified cilia reconstructed within the outline of the CNS from left lateral, dorsal, and frontal views. Cells coloured as in panels A-C, PR-I outer segments in yellow, Pr-II outer segments in teal, coronet bulbous protrusions in green.

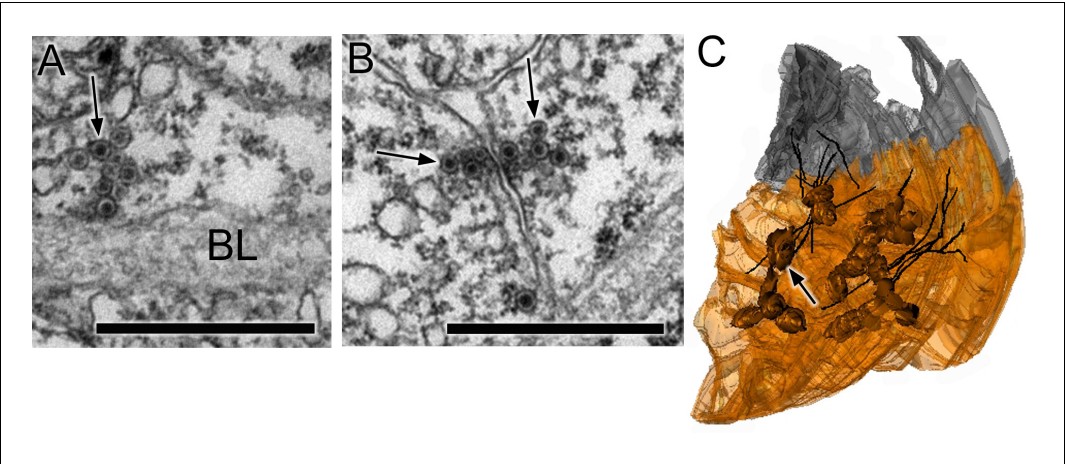

**Figure 6.** Synapses of coronet cells. (**A**) Synapse, containing exclusively dense-core vesicles (arrow), from a coronet cell onto the basal lamina (BL). (**B**) Unpolarized synapse between two coronet cells, with dense-core vesicles (arrows) on both sides of a synaptic cleft. (**C**) Reconstruction of coronet cells each with a bulbous protrusion (arrow) alongside coronet-cell associated somata (grey) of ciliated neurons, with cilia reconstructed in black. Scale bars: 1 μm (in A and B).

## Sidedness in the brain vesicle

Neurons are distributed unequally with respect to cell type. The most obvious case for sidedness in the CNS has been long recognized from the composition and placement of the ocellus and coronet cells (*Meinertzhagen and Okamura, 2001*; *Meinertzhagen et al., 2004*).

On the left side, we found the 17 enigmatic coronet cells (*Figure 5A–B*), each structurally distinguished by a single bulbous protrusion and expressing immunoreactivity to dopamine (*Moret et al., 2005*). These have short axons that

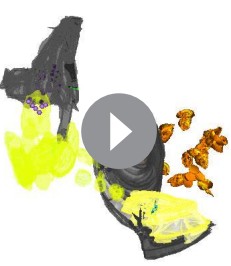

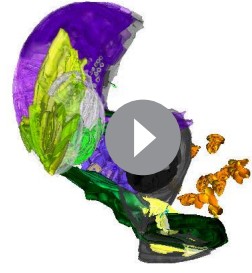

**Video 2.** Rotation of reconstructed sensory structures. Reconstructed pigment cells (black) with otolith associated ciliated cells (yellow) and vacuoles observed in a variety of cell types (lime green). Outer segments reconstructed as spheres for group I photoreceptors projecting into the ocellus (darker purple) and group II photoreceptors projecting into the canal (lighter purple) and group III modified outer segments (blue), as well as coronet bulbous protrusions (orange).

**Video 3.** Rotation of reconstructed sensory neurons. Reconstruction including transparent cell bodies illustrating pigment cells (black), group I (dark purple), group II (light purple) and group III (blue) photoreceptors with their outer segments reconstructed as spheres, lens cells (white) with their vacuoles reconstructed (bright green); also shown are vacuolated photoreceptor tract interneurons (lime), antenna cells (green), otolith associated ciliated cells (yellow), and coronet bulbous protrusions. Terminals of antenna and photoreceptor neurons are truncated in this view.

terminate against the basal lamina where they form synapses with almost exclusively dense-core vesicles 80–110 nm in diameter (*Figure 6A*). Additional synapses are formed at close range onto other interneurons, including relay neurons (pr-corRN and ant-corRN), or onto neighbouring coronet cells (*Figure 6B*). Alongside the coronet cells lie ciliated neurons, some of which have cilia that project towards the bulbous protrusions (*Figure 6C*).

On the right side, five rows of photoreceptors which, like those of vertebrates, and as previously reported (*Dilly, 1962*, *1969*; *Eakin and Kuda, 1971*), extend a stalk that contains a basal body and expands to form a ciliary outer segment in the ocellus pigment cup. In addition to the Group I photoreceptors of the ocellus, two anterior rows of Group II photoreceptors adjacent to the ocellus have ciliary outer segments that extend into the neural canal, with smaller terminals extending a short distance into the posterior brain vesicle neuropil. The number and arrangement of photoreceptor neurons confirms the presence of the additional class of photoreceptors identified using Ci-opsin immunolabeling, with outer segments outside the ocellus pigment (*Horie et al., 2005*, *2008*). These photoreceptors, along with those of the ocellus, total 30, as reported previously (*Horie et al., 2005*, *2008*). We identify 23 with outer segments that projected into the ocellus pigment, and 7 into the neural canal (*Video 2*; *Video 3*; *Video 4*). The 17–18 photoreceptors reported by *Nicol and Meinertzhagen (1991)* appear to represent just some of the rows of nuclei from the five fans of ocellus photoreceptors, those that were visible in semithin sections and that lay close to the opaque pigment. In addition to the 30 photoreceptors, three anterior sensory interneurons, unique in having axons within the sensory axon tract, lie on the right side in the anterior brain vesicle (*Figure 5C*). We also identify a further group of seven Group III (*Horie et al., 2008*) right-side photoreceptors which lie posterior and ventral to the 30 (*Figure 5C*), are vacuolated, and have outer segments that are less well organized.

## Sidedness in CNS pathways

In addition to receptor neurons, interneurons of the brain vesicle also exhibit sidedness in the position of their somata (*Figure 7*).

Right-side interneurons can be identified by various morphological features not resolved in previous light microscopy studies. Two classes of intrinsic interneurons are:

a. two bipolar neurons (cells 90 and 92), both photoreceptor interneurons (*Figure 8A*).
b. three anaxonal arborizing interneurons (aaINs), which receive inputs from antenna and photoreceptor neurons, with highly branched terminals adjacent to their cell bodies (*Figure 8B*).

Posterior to these intrinsic neurons are three additional classes of interneuron:

a. Teardrop-shaped neurons (cells 108, 116, 127, 157, 124, and 140) in the dorsal right brain vesicle that project axons ventrally to terminate in the anterior motor ganglion, one type with forked terminals (*Figure 8C*). This entire group also forms a connectivity class (pr-AMG relay neurons) that integrates input from photoreceptors and ascending peripheral interneurons (AMGs).
b. Two relay neurons (cells 123 and 130) that lack unique morphological features, but share aspects of their connectivity, both postsynaptic to photoreceptor and bipolar tail neurons.
c. Two antenna relay neurons (cells 142 and 152), one having an axon that splits to form long collateral bifid axons terminating at different depths in the motor ganglion, and the other having a long dendrite extending anteriorly from its soma (*Figure 8D,E*).

Most interneurons of the brain vesicle are however located on the left side and many are structurally anonymous. Aside from those mentioned above (a-c), photoreceptor and antenna relay neurons are left-sided. In addition, other relay interneurons that lack direct input from sensory neurons are also left-sided (*Table 3*).

The only apparent difference in photoreceptor input to left and right subclasses of relay neurons is that Type II canal photoreceptors are presynaptic only to right-side relay neurons of the pr-AMG class (*Figure 9*). Unlike their right-side counterparts PR-AMGRN(R) (cells 108, 116, 127, 157, 123, and 130), the left-side photoreceptor relay neurons PRRN(L) (cells 74, 94, 80, 86, 96, 100, 121, and 126) receive photoreceptor input exclusively from Type I ocellus photoreceptors (*Figure 9*). Two ventral antenna neurons (Ant1 and Ant2), which are proposed to signal input from otolith position (*Torrence, 1986*; *Tsuda et al., 2003*), lack obvious sidedness in their cell body positions, and extend collateral axons toward the posterior brain vesicle (BV). The terminals of these antenna cells,

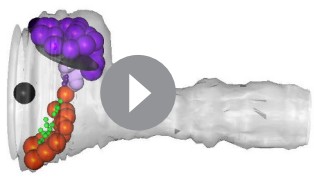

**Video 4.** Reconstruction of spheroids representing the cell body positions of sensory structures. Pigment, photoreceptor and coronet cells with the bulbous protrusions (green) and photoreceptor outer segments (type I and type II: yellow; and type III: purple).

however, differ in their synaptic input to sides of the posterior BV: Antenna cell 1 to both sides, and Antenna cell 2 predominantly to right-side relay neurons (AntRN). One right-side antenna relay neuron, located more ventrally (Ant-corRN; *Figure 1—source data 1*), is also postsynaptic to coronet cells.

Compared with the brain vesicle, the motor ganglion has paired neurons and thus appears more bilaterally symmetrical (*Figure 7*; *Figure 10—figure supplement 1*), but nevertheless receives asymmetrical inputs from the relay neurons of the brain vesicle (*Figure 10*). These project mostly to the motor ganglion's interneurons, however, although some sparse connections are made directly to the motor neurons themselves (*Figure 10—figure supplement 1*). The shortest sensory pathway to any motor neuron connects via a brain vesicle interneuron, and is thus disynaptic, although most direct pathways involve two interneurons and are thus trisynaptic (*Figure 11*). However, these shortest paths fail to depict the complexity of integration revealed in the total network (*Figure 7—figure supplement 1*; *Figure 9—figure supplement 1*).

The relay neurons of the photoreceptor and antennal pathways project asymmetrically (*Figure 10—source data 1*; *Figure 10—figure supplement 2*), photoreceptor interneurons connecting 70% to left-side interneurons, and antenna interneurons connecting 70% to right-side interneurons, of the motor ganglion (*Figure 10*; *Figure 10—source data 1*). The asymmetry is greatest among relay interneurons that receive additional input from coronet cells. Those that receive photoreceptor combined with coronet cell input project entirely to the left, and those that receive antennal and coronet cell input project entirely to the right. The asymmetry in pathways to motor ganglion interneurons parallels that of the direct input to motor neurons (photoreceptor pathways connecting to the left and antenna to the right) (*Figure 10*; *Figure 10—source data 1*).

## The motor ganglion

The five pairs of motor neurons (MN1-MN5) form morphologically distinct neuromuscular junctions. Two anterior primary motor neurons (MN1 and MN2) each form 200–360 large synapses with many vesicles and postsynaptic densities on the muscle (*Figure 12A*), whereas the three more posterior motor neurons (MN3-MN5) each form 15–50 smaller neuromuscular junctions, often with fewer vesicles and less obvious muscle postsynaptic specializations (*Figure 12B*). The input to left and right muscle at these junctions is asymmetrical from all motor neuron pairs except the MN2 neuron pair, which provides symmetrical synaptic input in both number (46% to left and 54% to right) and total size (49% to left and 51% to right). Asymmetries in synapses observed between left and right muscle are not great, however, falling within a left:right ratio range between 60:40 and 40:60, MN1 having greater input to the right, and MN3-MN5 greater to the left (*Table 4*).

Motor neurons also form synapses with each other, with some further asymmetries between connections on the left and right sides (*Figure 13A*). On the right side only, MN1 is presynaptic to MN4 and postsynaptic to MN3. MN2 also shows asymmetries, receiving input from MN3 on the right, but providing input to MN3 of the left, and receiving feedback input from MN5 just on the left side. These synaptic asymmetries contrast with the obvious symmetry of the network of gap junctions between the motor neurons (*Figure 13A–D*).

The synaptic strengths (see Materials and methods) of the interneuron network on left and right sides are also asymmetrical (*Figure 13E*). In particular, the strength of the synaptic input from the first interneuron (cell MGIN1) to the second (cell MGIN2) is far greater on the left side. This asymmetry is paralleled by the gap junction network, in which the second interneuron is connected to the first only on the left side. In addition, this second left interneuron (MGIN2L) is also connected to the contralateral first right interneuron (cell MGIN1R) by gap junctions (*Figure 13F*).

Sidedness in the neural network of the motor ganglion is also evident in the synaptic connections between interneurons and motor neurons (*Figure 14*). The first motor neuron receives more inputs

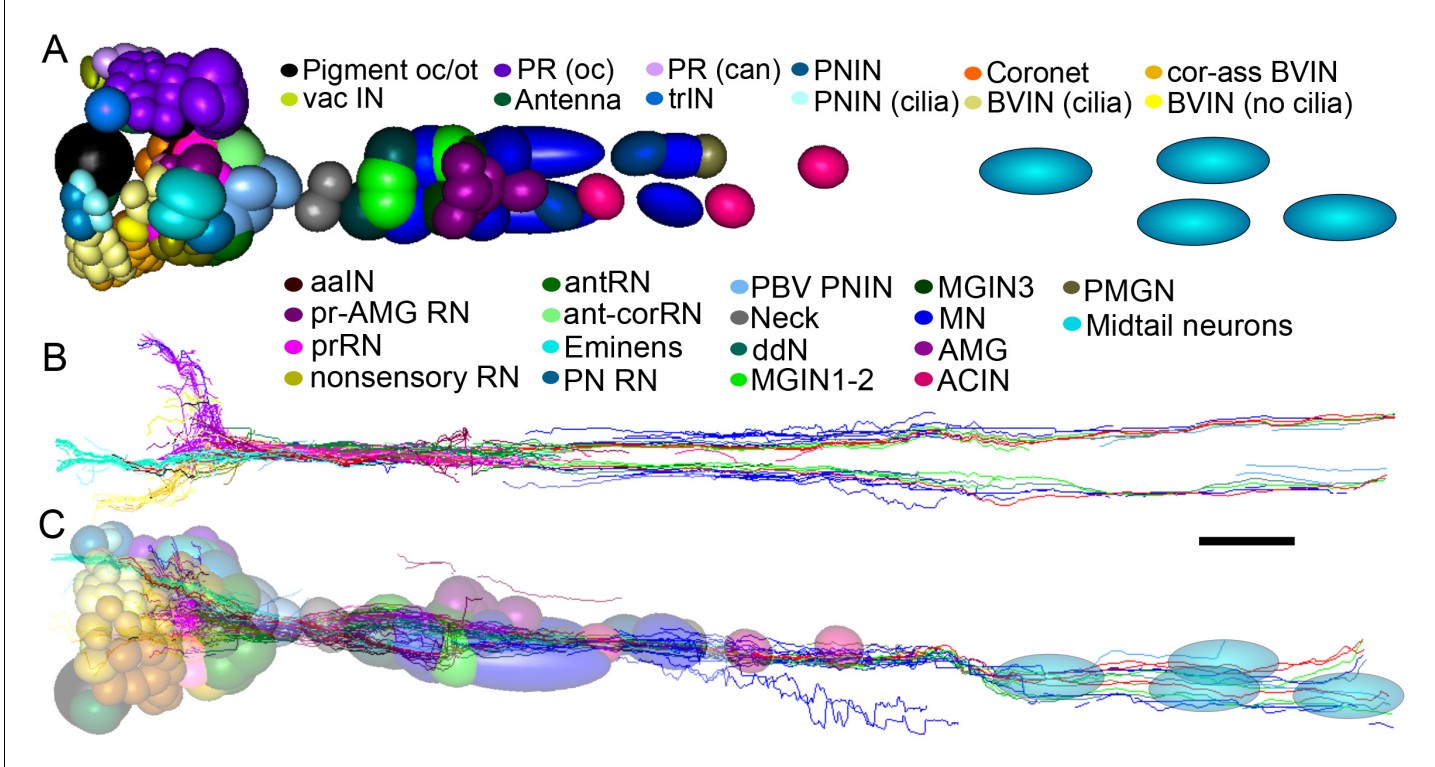

**Figure 7.** Representation and relative sizes of cell bodies and their positions along the neuraxis, with corresponding axon tracts. (A) Cell bodies of CNS neurons, dorsal view. Colours denote cell types (key). (B) Corresponding axon tracts, shown as skeleton reconstruction, dorsal view, colours as in (A) (for a network graph of synaptic connectome formed by corresponding neurons sorted by connectivity see *Figure 7—figure supplement 1*). (C) Cell bodies of CNS neurons and axon tracts, corresponding to (A) and (B), left lateral view. Pigment oc/ot: ocellus and otolith pigment cells; PR (oc): type I photoreceptor; PR (can): type II photoreceptor; PNIN: peripheral interneuron; PNIN (cilia) peripheral interneuron with cilium; vac IN: vacuolated photoreceptor-associated interneuron; Antenna: antenna cell; Coronet: coronet cell; aaIN: anaxonal arborizing interneuron; BVIN (cilia): ciliated brain vesicle interneuron: pr-AMG RN: photoreceptor-AMG relay neuron; trIN: photoreceptor tract interneuron; cor-ass BVIN: ciliated coronet associated brain vesicle interneuron; prRN: photoreceptor relay neuron; BVIN (no cilia): brain vesicle interneuron lacking cilium; non-sensory relay neuron (RN); antRN: antenna relay neuron; ant-corRN: antenna-coronet relay neuron: Eminens: eminens neuron: PNRN: peripheral relay neuron: PBV PNIN: posterior brain vesicle peripheral interneuron; MGINs 1–2: motor ganglion paired interneurons 1 and 2; MGINs 3: motor ganglion paired interneurons 3; ddN: descending decussating neuron pair: AMG: ascending motor ganglion neuron; MN: motor neuron: ACIN: ascending contralateral inhibitory neuron; PMGN: posterior motor ganglion neuron; Midtail neurons: short descending neurons of the caudal nerve cord. Scale bar 10 μm.

The following figure supplement is available for figure 7:

**Figure supplement 1.** Total network of synaptic pathways within the larval CNS of *Ciona intestinalis*.

on the right side from both the first and third interneuron (cells MGIN1 and MGIN3) than it does on the left (synaptic depths 1.08 and 1.2 μm versus 0.42 and 0.72 μm). The second motor neuron (MN2) likewise receives more inputs from the second interneuron on the right (MGIN2R) than the left (MGIN2L) side (synaptic depths 4.69 versus 1.24 μm). Other asymmetries are apparent in connections between cells that are reciprocal on only one side: MN1L to MGIN3L, MN3L to MGIN2L, on the left, and MN5R to MGIN1R, MN1R to MGIN2R, and MGIN2R to MN4R on the right (*Figure 14A*). These pathways refer to chemical synapses (*Figure 14A*), relative to which those exhibited by gap junctions (*Figure 14B*) are more left/right symmetrical and involve more connected partners across the midline, especially for the anterior components.

Aside from the obvious asymmetry in their number, ACINs are also asymmetric in their projections and connections between left and right sides of the posterior motor ganglion (*Video 5*). On the right, although the ACIN crosses the midline, it does not pass the ependymal cell to extend into the right neuropil, and so forms no contralateral synapses. On the left, however, both ACINs decussate fully, crossing to the contralateral neuropil and forming synapses there. These synapses are as a

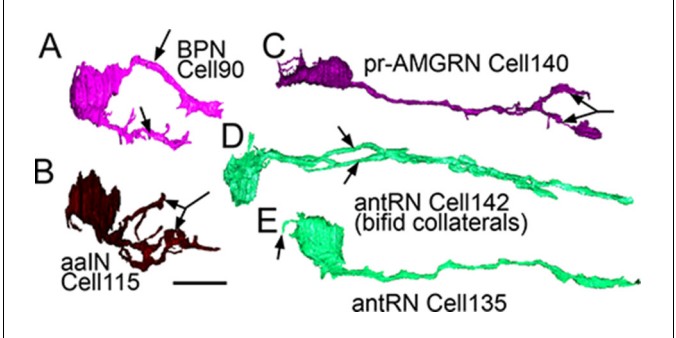

**Figure 8.** Right-side interneurons reconstructed from the brain vesicle, left lateral views, anterior to the left. (A) Intrinsic bipolar interneuron with two axons (arrows). (B) Anaxonal arborizing interneuron with large branched terminal (arrow). (C) Photoreceptor-ascending motor ganglion (pr-AMGRN) relay neuron with forked terminal (arrow). (D) Antenna relay neuron with bifid axon (arrows). (E) Antenna relay neuron with single axon, terminal and soma dendrite (arrow). Scale bar 10 μm.

result not formed directly onto contralateral motor neurons, as previously proposed (*Horie et al., 2010*), but instead onto interneurons (cells MGIN1R, MGIN2R and MGIN3R) of the right side (*Figure 15A*). Both left and right ACINs are presynaptic to their ipsilateral motor neurons (*Figure 15B*) and first two pairs of interneurons. All three ACINs are also presynaptic to a right-side bipolar tail neuron (cell BTN2, reported in *Stolfi et al., 2015*), and additionally form multiple synaptic contacts directly onto the ventral basal lamina (*Figure 15C*). That input resembles the synaptic input from the motor neuron terminals, which form onto the basal lamina that ensheathes the muscle (*Sanes et al., 1978*).

Presynaptic input to ACINs is not only asymmetric, but differs between each of the three ACIN cells. All are postsynaptic, however, to at least one ipsilateral interneuron and one ipsilateral motor neuron (*Figure 15D*). Both left ACINs receive input from MN3L and MGIN2L, but ACIN1L receives additional input from MN1L and a descending decussating neuron (cell ddR), while ACIN2L receives input from MGIN1L and the third a contralateral interneuron (MGIN3R). On the right, the only symmetrical input partner is the ipsilateral first interneuron, MGIN1(R), with asymmetrical input from MN2R as well as right-side bipolar tail neurons.

Summarizing: the connectivity matrix of all neurons reveals the following features (*Figure 16*): (a) Most cell types form synapses among members of the same class of neuron. (b) Many synapses also form on the basal lamina and from motor neurons onto muscle cells. (c) A high degree of reciprocity is manifest between members of different neuron classes, especially those that are interneurons. (d) Neurons are both pre- and postsynaptic, those that have many presynaptic sites generally also have many that are postsynaptic, except for sensory neurons in which presynaptic sites predominate. (e) The pathway strength is in general greater in the motor ganglion but also high for the relay neurons and among neurons of the pathway from the peripheral neurons. (f) Cell classes with fewer representative neurons tend to have greater pathway strengths. Features of the sidedness of the matrix are reported in *Figure 16—figure supplement 1*

## Discussion

We report the full synaptic connectome of a single tadpole larva of a model chordate species, the ascidian *Ciona intestinalis*, and use this to identify the complete inventory of the many asymmetrical features in its CNS. Some connectomic analyses rely on symmetry of the connections between left and right sides to validate the synaptic connections of neurons that are paired (*Durbin, 1987*; *Randel et al., 2014*; *Ohyama et al., 2015*). This approach has not been possible in *Ciona*, because many cells are not bilaterally paired in the brain vesicle, while in the motor ganglion where cells are paired, presynaptic inputs from the brain vesicle are similarly asymmetrical. Additionally, we cannot exclude the possibility that minor left/right asymmetries might be the product of developmental

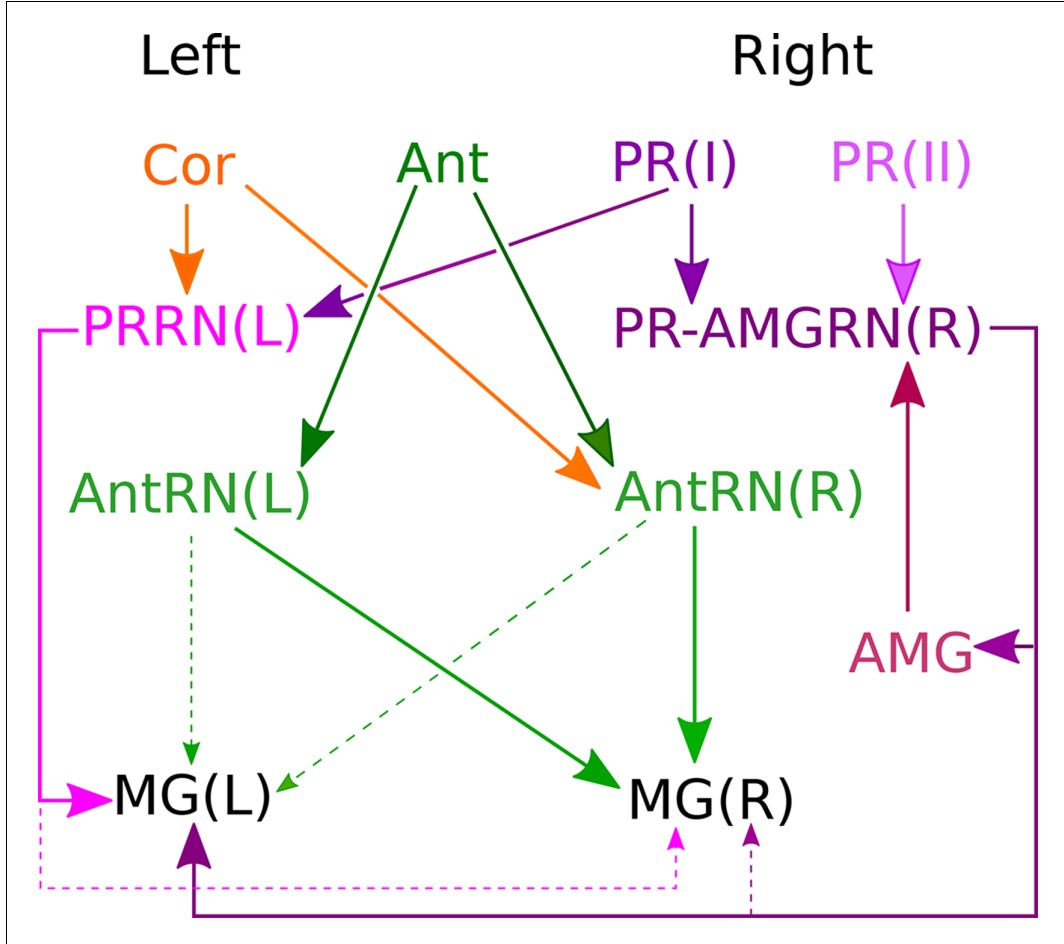

**Figure 9.** Asymmetrical sensory input to the two sides of the motor ganglion MG(L) and MG(R) via relay neurons. Sensory input arises from coronet cells (Cor); antenna cells Ant1 and Ant2 (combined as Ant); and photoreceptors (PR) of two types: ocelli (oc: PR I) and neural canal (can: PR II). Signals are relayed through respective interneuron classes: photoreceptor relay neurons on the left, PRRN(L); photoreceptor-ascending motor ganglion (PR-AMGRN (R)) relay neurons on the right (*Figure 8c*); and antenna relay neurons (AntRN) of the left and right sides. PR-AMG relay neurons of the right side receive input from ascending motor ganglion neurons that is reciprocated. Pathways with weak connections are shown with dashed lines. Details of pathway strength appear in *Figure 9— figure supplement 1*

The following figure supplement is available for figure 9:

**Figure supplement 1.** Total network of synaptic pathways within the larval CNS of *Ciona intestinalis*.

noise or imprecision, insofar as only a small number of larvae need to survive and eventually become adults.

Our findings reveal not only those features with possible counterparts in the vertebrate CNS, but also the many features of all nervous systems. These include the wide range and combination of cells that share synaptic contacts, the poorly segregated distribution of synapses over the neuron surface; synapses onto non-neuronal cells and basal lamina; unpolarized and mixed vesicle synapses resembling those in cnidarians (*Westfall, 1996*); and the apparent general redundancy in the connectome. Unpolarized synapses have been previously reported between coelenterate (*Horridge and Mackay, 1962*) and pulmonate mollusc (*McCarrager and Chase, 1985*) neurons, while synapses onto the basal lamina have been reported in muscle after removing the underlying myofibres (*Sanes et al., 1978*). Synaptic sites that occur onto the basal lamina, and that therefore lack postsynaptic partners, resemble other presynaptic sites in the CNS, and resemble neuromuscular junctions, where

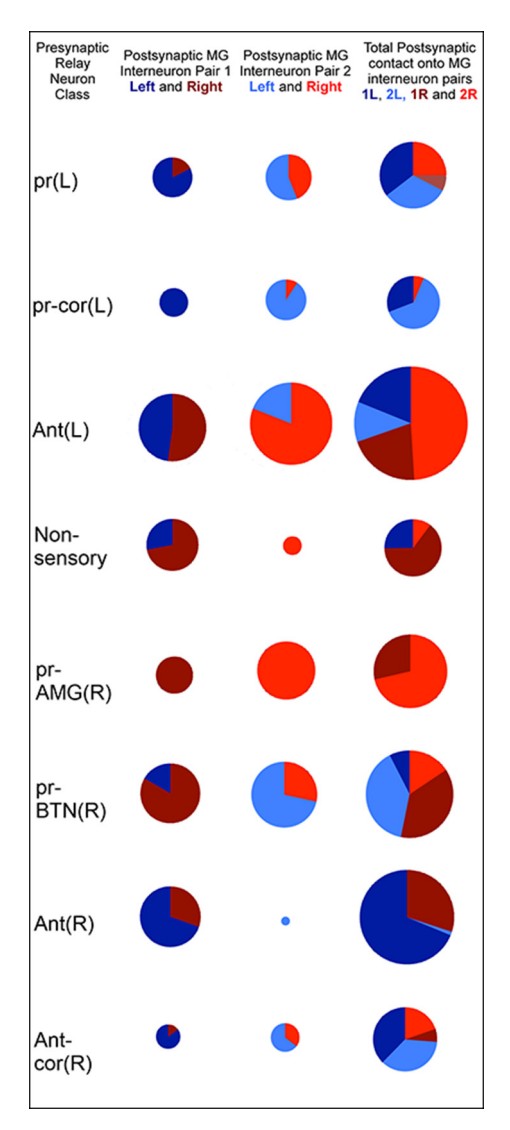

**Figure 10.** Classes of relay neurons (presynaptic) in the CNS of *Ciona* and the inputs these provide to cells on the left and right sides of the motor ganglion (for details of relay inputs see *Figure 10—figure supplement 1* and for antenna pathway see *Figure 10—figure supplement 2*). For relay neuron class names see the key in *Figure 1—source data 1*). Each circle represents the input synapses to the first (column 1), second (column 2) or both (column 3) paired MG interneurons. Inputs to the left (blue) or right (red) partners are shown as an angular subtense of a circle the area of which represents the overall synaptic strength. Most sensory inputs are predominantly one-sided, some entirely so. Total synaptic input varies widely (see *Figure 10—source data 1* for actual values and proportions).

The following source data and figure supplements are available for figure 10:

**Source data 1.** Relay neuron inputs to the left and right motor ganglion.

**Figure supplement 1.** Reconstructions of motor ganglion neurons populated with photoreceptor and antenna relay neuron synaptic input sites, colour-coded by relay neuron type (key).

**Figure supplement 2.** Antenna cell relay neuron input to the motor ganglion.

neurotransmitter must cross the basal lamina to act on adjacent muscle cells. Some presumed sites of synaptic release across the basal lamina lie opposite non-neuronal cells, including epidermal cells or cells of the notochord. Neurotransmitter receptor expression studies are critical to our understanding of the roles these special sites of synaptic vesicle release may play. Sites that lack adjacent cells may be sites of neuromodulator release. This is particularly likely in the case of coronet cells, which express reporters for dopamine and have exclusively dense-core vesicles at their synapses onto the basal lamina. Synaptic reciprocity and serial networks are commonplace in many nervous systems (e.g. *Dowling and Boycott, 1966*). Aided by the numerical simplicity of the *Ciona* CNS, we are also able to detect features such as cilia that may lie undetected in highly populated vertebrate brains, and confirm that many neurons in *Ciona* are ciliated (*Figure 3—figure supplement 1*).

An obvious point of comparison is set by the nervous system of *C. elegans*, the sole precedent for a completely reported connectome, with which the *Ciona* larva is dimensionally comparable and with which it has a numerically similar network – 302 identified neurons in the nervous system of a single hermaphrodite *C. elegans* (*White et al., 1986*; *Varshney et al., 2011*) compared with 177 neurons in the larval CNS of *Ciona*. Neurons form comparable numbers of synapses in both these model nervous systems – an average of 37 presynaptic sites and 7 putative gap junctions (>1 section) compared with 28 synapses and 3.2 gap junctions per non-pharyngeal neuron in *C. elegans* (calculated from data at http://www.wormatlas.org/hermaphrodite/nervous/Neuroframeset.html). These numbers are smaller than in *Drosophila*, in which optic lobe neurons may have in excess of 100 presynaptic sites (*Meinertzhagen and Sorra, 2001*; *Takemura et al., 2008*, *2015*) and clearly less than typical vertebrate neurons (for example a mouse somatosensory cortex neuron has about 8200 synapses: *Schüz and Palm, 1989*). Further discussion of this topic is provided elsewhere (*Meinertzhagen, 2010*).

For brain asymmetry to appear in the hatched larva, the expression of Nodal and ciliary action are both required during embryonic development (*Nishide et al., 2012*; *Thompson et al., 2012*). These events are perturbed during the process of dechorionation that has been used for many experimental interventions, especially electroporation of genetic reagents (*Shimeld and Levin, 2006*), so that many asymmetries may have escaped detection in previous reports that are revealed in our larva reared with an intact chorion. Furthermore, within the chorion the developing embryo invariably curls around itself along the left side of the trunk (*Katsumoto et al., 2013*), and this may further influence sidedness in the larval brain, in ways that are lost after dechorionation. In addition to pigment cell displacement, arrestin expression indicating photoreceptor cell fate, is significantly altered, often expanding to the left brain in dechorionated embryos (*Oonuma et al., 2016*).

The overall asymmetry of the larval ascidian CNS finds deep parallels with asymmetries in other chordates, including vertebrates (*Boorman and Shimeld, 2002b*), and clear differences from those of other deuterostomes, such as echinoderms (*Duboc et al., 2005*). Most difficulties in comparing between ascidian larval and vertebrate nervous systems come however from differences in their respective cell numbers. The extreme miniaturization of the former exposes sidedness in the ascidian larval brain that can actually exist in both chordate clades, such as in the vertebrate epithalamus (*Concha and Wilson, 2001*; *Hamada et al., 2002*). Structurally most obvious in the ascidian larva is sidedness in the position of the ocellus, which is driven by the same left-side action of Nodal (*Yoshida and Saiga, 2011*) as drives asymmetry in all chordate brains, including those of vertebrates (*Halpern, 2003*; *Carl et al., 2007*).

Migration of the ocellus pigment driven in *Ciona* by Nodal (*Yoshida and Saiga, 2011*) calls to mind the vertebrate pineal, which is also lateralized based on Nodal's action (*Carl et al., 2007*). Nodal acts on the left side of the CNS in both, but in *Ciona* the pigment cells migrate to the right, whereas vertebrate parapineal cells migrate to the left. Moreover, based on the projection of their outer segments, which is inward, not toward the outside world, *Ciona*'s larval eyes are more akin to lateral eyes than to a single pineal (*Lamb, 2013*). The photoreceptor cells of the ocellus likely share a common lineage, blastomeres A9.14 and A9.16 on the right side (*Oonuma et al., 2016*), with the coronet cells of the left side (*Cole and Meinertzhagen, 2004*). Ascidian coronet cells are likewise components of a morphologically homologous structure that is bilateral in vertebrates, the saccus vasculosus (*Smeets et al., 1983*). In vertebrates, the saccus vasculosus comprises coronet cells in addition to neurons contacting the cerebrospinal fluid. In *Ciona* we report new ciliated coronet-associated neurons, with cilia that project into the neural canal toward the coronet cells' bulbous

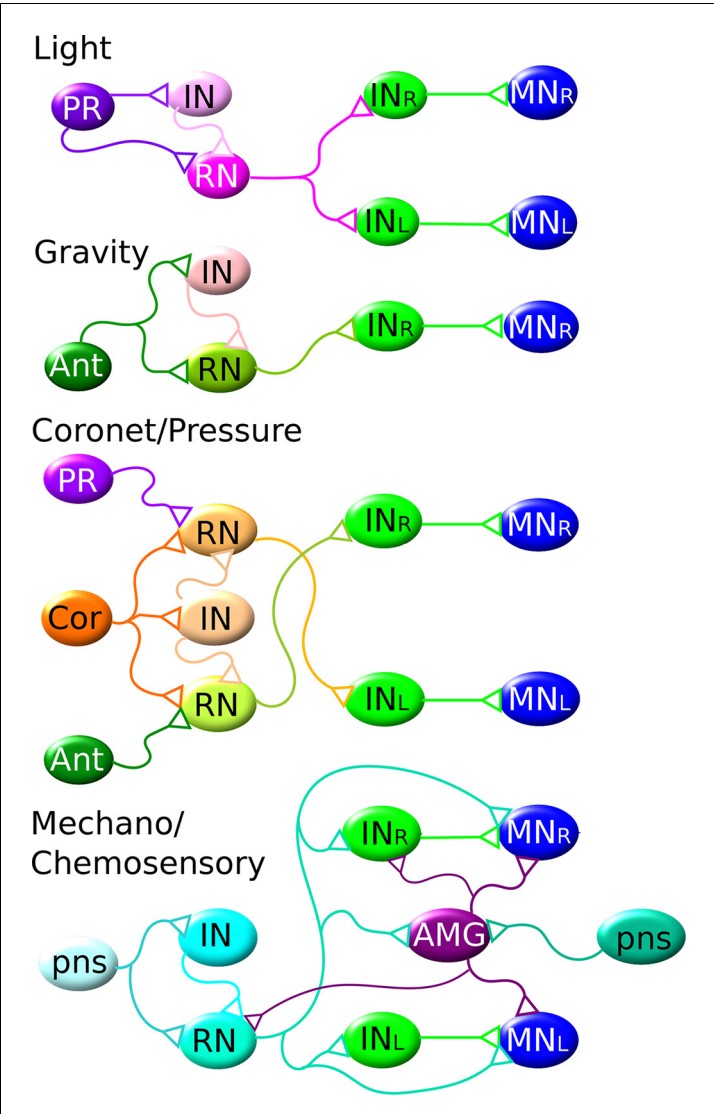

**Figure 11.** The shortest CNS pathways between sensory neurons and motor neurons for different sensory modalities are three-synapse arcs. Four modalities are indicated, from top to bottom: light, gravity, coronet cells (possibly hydrostatic pressure) and PNS mechano/chemosensory. Members of the same cell types are assigned the same colour. Each pathway originates in the particular class of sensory neuron (photoreceptor: PR; antenna neuron: Ant; coronet cell, Cor; and PNS sensory neuron, pns) connecting via relay neurons (RN) and interneurons (IN) of the brain vesicle, to interneurons (IN) of the motor ganglion, to motor neurons (MNs). AMG: addition relay neurons of the motor ganglion receive input from the PNS. These pathways are all interconnected, overlapping networks for all sensory modalities thus underlying the complex reality of the actual behavioral network (*Figure 9—figure supplement 1*).

protrusions. In both vertebrate and ascidian cases, it is these neurons rather than the coronet cells that give rise to an axonal pathway, forming the axon tract emanating from the saccus vasculosus in vertebrates (*Rodríguez-Moldes and Anadón, 1988*) and the coronet complex in *Ciona*, further strengthening the similarities between coronet complex and saccus vasculosus. Both ocellus and coronet complex structures are lateralized in *Ciona*, unlike the single medial structures reported for cyclopic mutants (*Belloni et al., 1996*; *Chiang et al., 1996*) or fused, unpaired saccus vasculosus phenotypes (*Nieuwenhuys, 1998*).

Given that ascidian larvae swim in a helical pattern (*McHenry, 2005*) and have a single-sided ocellus, their phototactic behaviour follows a helical, not visual pattern, as defined by *Randel and Jékely*

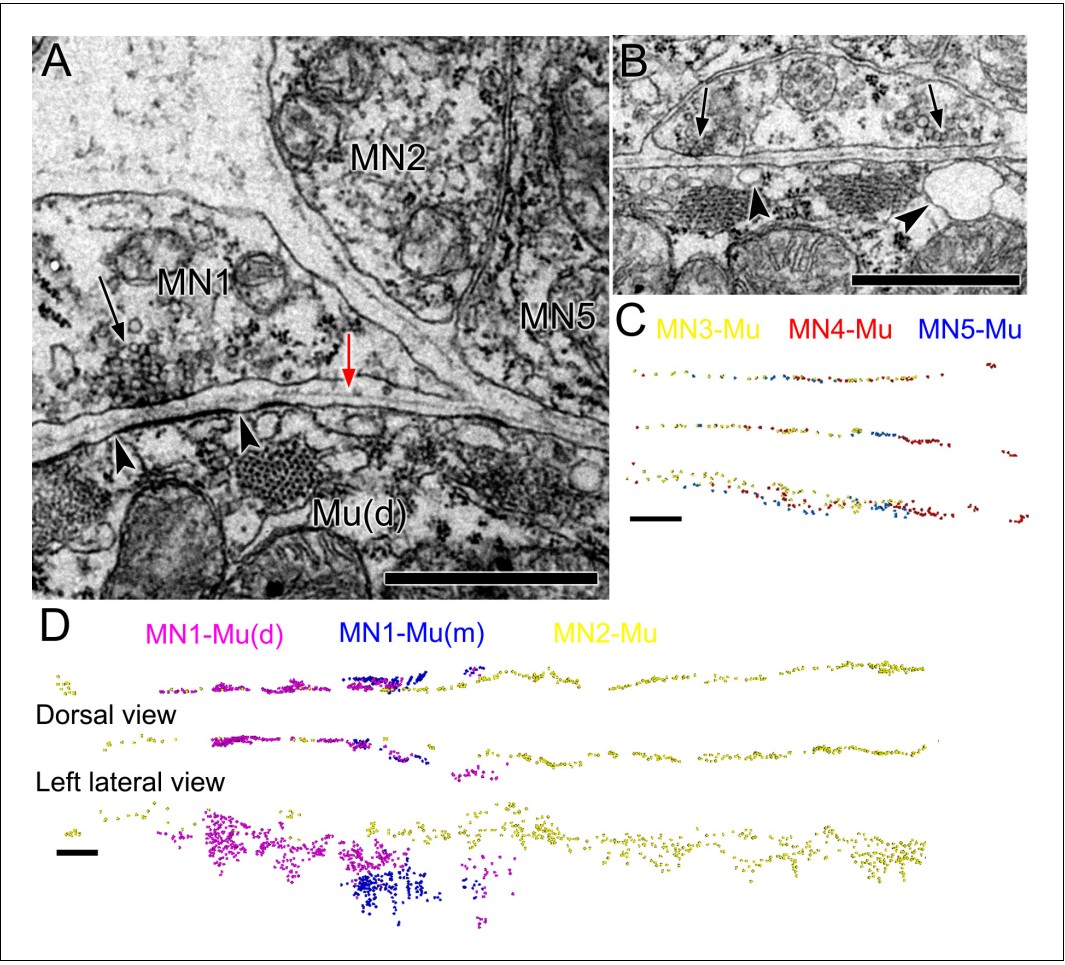

**Figure 12.** *Ciona intestinalis* larval motor neuron terminals and neuromuscular junctions. (**A**) Neuromuscular junction (arrow) of MN1, with a postsynaptic specialization on the muscle (arrowheads). A basal lamina (red arrow) extends in the cleft between neuron and muscle. (**B**) Two adjacent neuromuscular synapses (arrows) with postsynaptic cisternae (arrowheads), but lacking postsynaptic membrane densities. (**C,D**) Enlarged views of anterior tail with reconstructed puncta representing colour-coded neuromuscular junctions of each motor neuron pair. (**D**) Top: Dorsal view. Bottom: Left lateral view. Scale bars: 1 µm (**A, B**); 10 µm (**C, D**).

**Table 4.** Input to left and right dorsal and medial muscle bands from motor neuron pairs at their neuromuscular junctions.

| Motor neuron pair | Left muscle band | | | | Right muscle band | | | | Ratios | |
| | Dorsal | | Medial | | Dorsal | | Medial | | Left: Right | |
| | No. syn | No. sec | No. syn | No. sec | No. syn | No. sec | No. syn | No. sec | No. syn | No. sec |
|---|---|---|---|---|---|---|---|---|---|---|
| MN1 | 192 | 969 | 47 | 145 | 230 | 1181 | 130 | 558 | 40: 60 | 39: 61 |
| MN2 | 224 | 1583 | | | 258 | 1636 | | | 46: 54 | 49: 51 |
| MN3 | 42 | 156 | | | 28 | 101 | | | 60: 40 | 61: 39 |
| MN4 | 45 | 189 | | | 30 | 116 | | | 60: 40 | 62: 38 |
| MN5 | 21 | 128 | | | 15 | 55 | | | 58: 42 | 70: 30 |

Number of synapses (No. syn) and number of synaptic profiles (No. sec) provided for each motor neuron and left:right ratios expressed as percentages of neuromuscular junction input from left and right partners for each motor neuron pair.

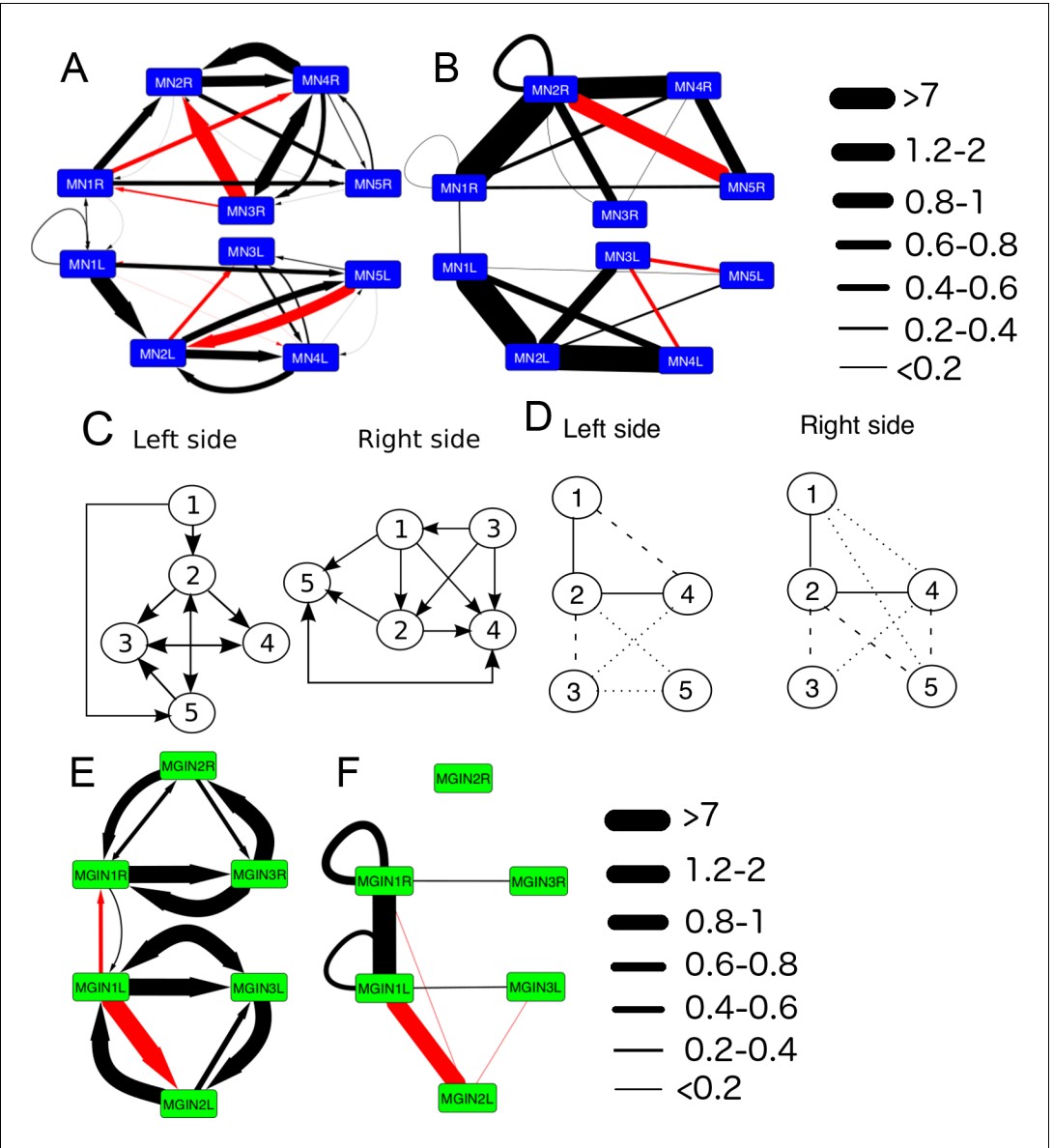

**Figure 13.** The networks of motor neurons MN1-MN5 and descending ipsilateral neurons (MG1-MG3) of the left and right side of the motor ganglion (MG). (**A**) Synaptic network of motor neurons 1–5 on the left (MN1-5L) and right (MN1-5R) sides. (**B**) Network of putative gap junctions between motor neurons of the MG. (**C**) Summary diagrams of motor neuron synaptic networks of left and right sides. (**D**) Summary diagrams of putative gap junction network of motor neurons of the left and right sides. Dotted lines represent tentative connections, dashed lines minimal contacts, and solid line connections with many large contact sites. (**E**) Synaptic network of descending ipsilateral interneurons (MGINs) of the motor ganglion. (**F**) Network of putative gap junctions between descending ipsilateral interneurons of the motor ganglion. For a, b, e and f: Arrows illustrate polarity of synapse, line thickness show the cumulative depth of synaptic contact in μm (see Materials and methods). Red lines illustrate synaptic contacts that differ between left and right sides of the MG.

*(2016)*. Unlike simple helical phototaxis, however, the complement of cell types and complexity of component connections involved in *Ciona*'s visual circuit (*Video 6*) compare much more closely to circuits responsible for visual phototaxis than those for simple helical phototaxis (*Randel and Jékely, 2016*: their *Figure 2f,g*). Helical swimming of the ascidian larva provides a mechanism by which a preexisting bilateral visual phototaxis circuit could have been co-opted into a complex hybrid helical phototaxis circuit, still allowing mechanisms such as delay, sensory integration, and modulation to

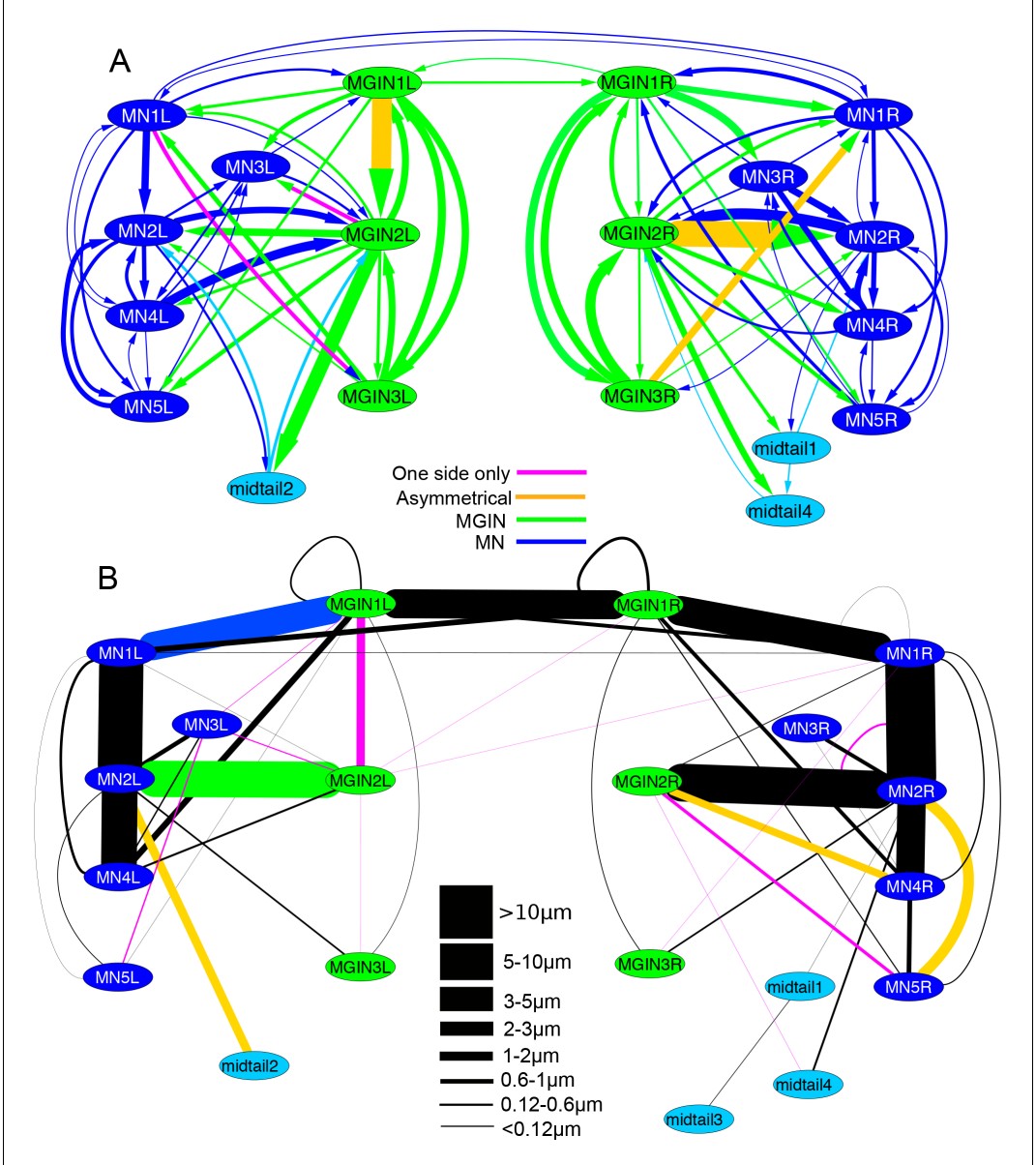

**Figure 14.** Left-right asymmetries in the overall synaptic pathways of the motor ganglion. Pathways shown are between motor neurons (MNs), descending ipsilateral interneurons (MGINs) and descending mid-tail neurons. (**A**) Synaptic network with arrows indicating polarity of synaptic contacts. (**B**) Summary network of gap junctions for descending motor ganglion neurons illustrated in (**A**). Blue lines represent gap junction inputs from motor neurons, green represent gap junction inputs from interneurons, and pale blue are gap junction inputs from mid-tail neurons. Pathway strength varies over a wide (>25 times) range and is more left-right symmetrical than the synaptic network in (**A**). In both (**A**) and (**B**) pathways shown by orange lines are left/right asymmetrical, and those in pink are present only on one side (key). Thickness of lines indicates cumulative depth of synaptic contacts (see Materials and methods) (scale).

take place, and unlike the more direct helical phototaxis mechanisms in ciliated forms, such as protists and trochophore larvae (*Randel and Jékely, 2016*).

The presence of unilateral systems in the larval CNS of *Ciona* that show homology with bilateral structures in the vertebrate brain can be interpreted as a reduction in one side, and as an outcome of the small cell numbers in ascidian larvae. A defining feature of larval *Ciona* and its CNS is indeed the miniaturization of both. The larva's small cell numbers can be seen as a direct outcome of the

few embryonic cleavage generations, no more than 14 for the entire CNS (*Cole and Meinertzhagen, 2004*), the lack of metamerism (*Garstang, 1928*; *Crowther and Whittaker, 1994*), the lack of feeding in this lecithotrophic larvae, and consequently its short life. The small number of its component neurons is reflected in turn in the relatively small numbers of synapses formed by each, conforming to an overall relationship between neuron number and synapse number per neuron seen in nervous systems generally. In contrast to both is the richness of cell types. Among the 177 larval neurons we can distinguish at least 25 different types and 52 subtypes identified on the basis of morphological and connectivity differences. *C. elegans* has a numerically comparable richness, with 118 cell types among its 302 neurons (*White et al., 1986*).

The asymmetries we observe start to appear with the failure of cells to pair left and right, at around 75% of embryonic development (*Cole and Meinertzhagen, 2004*). After this the brain vesicle pushes to the right between 75% and 85% of embryonic development, and cell positions begin to shift, followed by the loss of strict bilateral symmetry among the cells of the motor ganglion. The pattern of cell lineage is left/right symmetrical until the 11th cleavage, with ventral divisions becoming desynchronized after the 10th cleavage (*Cole and Meinertzhagen, 2004*). The photoreceptor pigment cell begins its migration at the 11th generation. The ACINs appear after the 11th

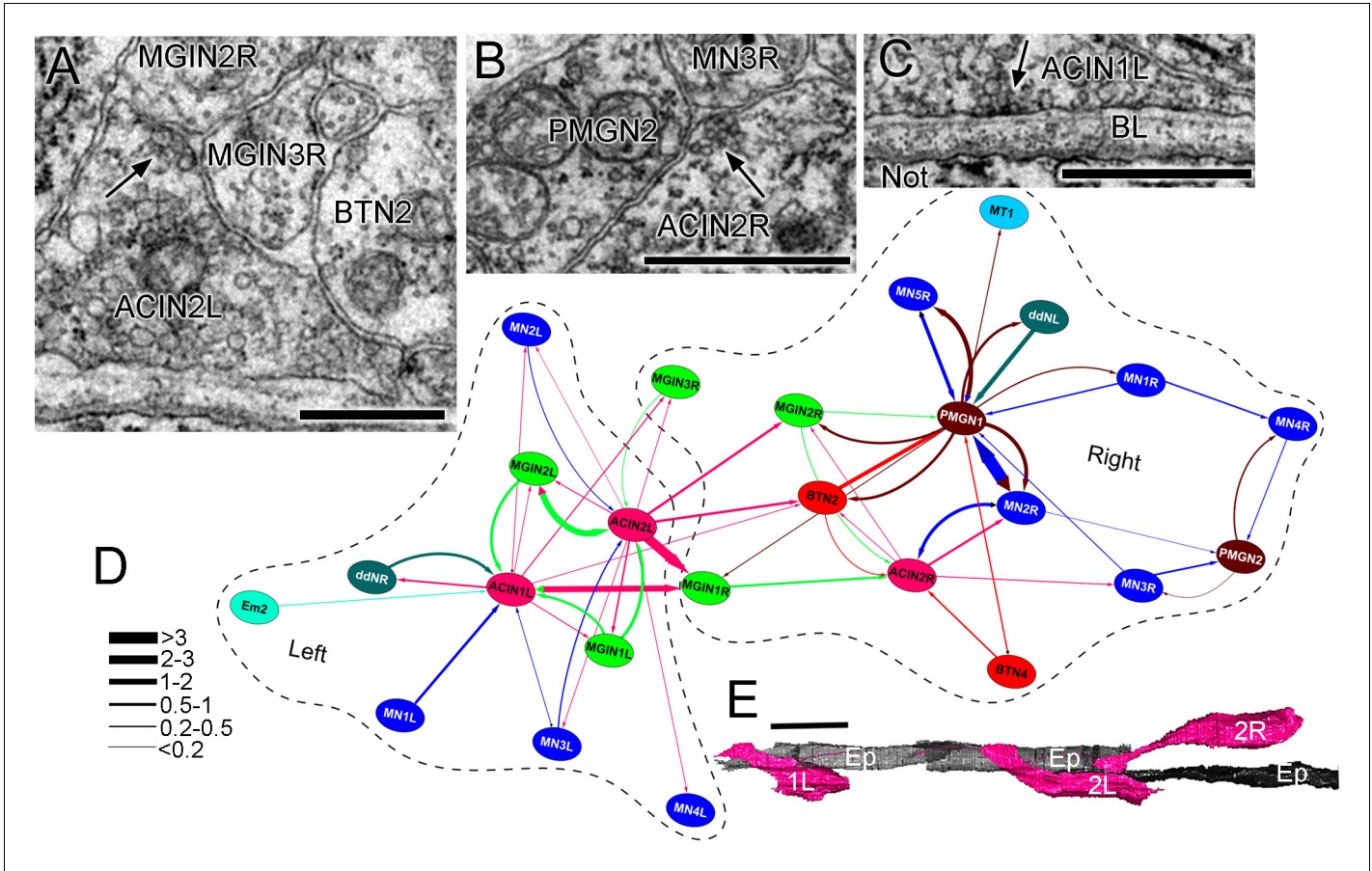

**Figure 15.** ACIN synapses and network. (**A**) Presynaptic site (arrow) from the left ACIN onto contralateral MGIN interneurons at a dyad synapse. BTN2: bipolar tail neuron profile. (**B**) Dyad synapse (arrow) onto ipsilateral motor neuron MN3R and an unpaired tail interneuron (PMGN2) on the right side. (**C**) Synapse (arrow) from ACIN1L onto the ventral basal lamina (BL) opposite the notochord. Scale bar (**a-c**): 1 μm. (**D**) Network diagram of ACIN pathways. Layout plotted as an edge-weighted spring embedded network (Cytoscape 3.1.0: NRNB.org) based on synapse pathway strengths (see Materials and methods). Right and left neuropiles are each enclosed in a dashed line. Pathway strengths are shown as the line thickness sorted by the cumulative depth of synaptic profiles (key). The right side includes two sided PMGN interneurons and their partners. Note reciprocity of connections for ipsilateral but not contralateral partners. Thus ACINs are presynaptic to contralateral partners but not postsynaptic. Cell types abbreviated as in *Figure 1—source data 1* (**E**) Dorsal view of reconstructed ACINs. Scale bar 10 μm.

**Video 5.** Rotations of reconstructed ACINs decorated with their presynaptic sites. Reconstructed ACINs with presynaptic sites colour-coded by postsynaptic cell type: basal lamina (black); motor neuron (blue), descending MG interneuron (green), bipolar tail neuron (red), and posterior MG descending neuron (brown)

embryonic cleavage (*Nishitsuji et al., 2012*) and fate choices select cells as either neurons or ependymal cells at around this stage too (*Cole and Meinertzhagen, 2004*). The asymmetry in position within the motor ganglion and between the cell numbers of the two sides of the caudal nerve cord is apparent by the final stages of embryonic development.

Total cell numbers are mostly left-right symmetrical throughout the CNS, and sidedness is mostly a question of cell fate. Thus the fate choice, for example between neuron and ependymal cell type, generates the sidedness in the numbers of cells comprising the same class. A striking example is provided by the ACIN neurons in the rostral caudal nerve cord. Thus A11.116 undergoes two divisions, first to yield A12.231 and A12.232 (*Nishitsuji et al., 2012*: their *Figure 5B*), and a further round of divisions, yielding a total of four 13th-generation cells that become two ACIN and two ependymal cells. The fate decision to generate ependymal and ACIN cells is then proposed to occur after this final division, based on the expression of extracellular signals and transcription factors. The total number of ACINs varies between larvae (*Nishino et al., 2010*), and in the larva we report, this variation manifests itself as an asymmetry, in which we observe two progeny of the left side as ACINs, with only a single ACIN present on the right, the other site being occupied by ependymal cell tail 4. Thus variation in cell fate decision is, we propose, the basis for the left/right asymmetry in ACIN neurons. We also find neurons on the right side (PMGN1 and PMGN2) of a hitherto unreported type that are located anterior to the single ACIN (ACIN2L) of that side, but posterior to motor neuron pair MN5. Their asymmetrical location suggests they may represent similar late choices between neuronal and ependymal cell fate.

Together with these asymmetries in cell types, there are clear asymmetries in connectivity. These are most obvious in inputs to, and connections within, the motor ganglion. Similar asymmetries may exist in other motor systems but are revealed only from comparisons between the left and right sides of a complete network, and so have rarely been revealed. In the mouse, however, an imbalance index for the motor innervation of interscutularis muscles reveals an asymmetry in the morphological features of motor innervation (*Lu et al., 2009*), while in the polychaete *Platynereis* there is also an asymmetric connectivity pattern for one class of motor neuron, both for the inputs it provides to the muscle it innervates and the inputs it provides to contralateral neuron partners (*Randel et al., 2014*, *2015*). These instances compare to both the asymmetrical input of, for example, MN1 to muscle and other network asymmetries we find in *Ciona* in which, like *Platynereis*, an input from two ACINs on the left side of the brain to interneurons on the right side is not matched by a contralateral input from the ACIN on the right side.

Our conclusions are based on a single larva, although four unrelated sibling larvae have been reported with closely similar overall cell complements (*Nicol and Meinertzhagen, 1991*). Data are still lacking on the constancy of neuron cell types and their connections between individuals. The consistency of asymmetrical differences we see in this larva will only be resolved when its sibling larvae are examined, as we now undertake.

Mechanosensory drive of symmetrical swimming was initially dismissed because tails can retain swimming independent of the head, and no known input from mechanosensory cells to motor neurons was known. However, we find multiple inputs from mechanosensory tail

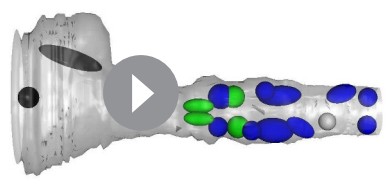

**Video 6.** Animated reconstruction of the photoreceptor pathway. Cell bodies, shown as spheroids, from photoreceptor through relay neurons to the motor ganglion. Cells are colour-coded as in *Figure 1—source data 1*

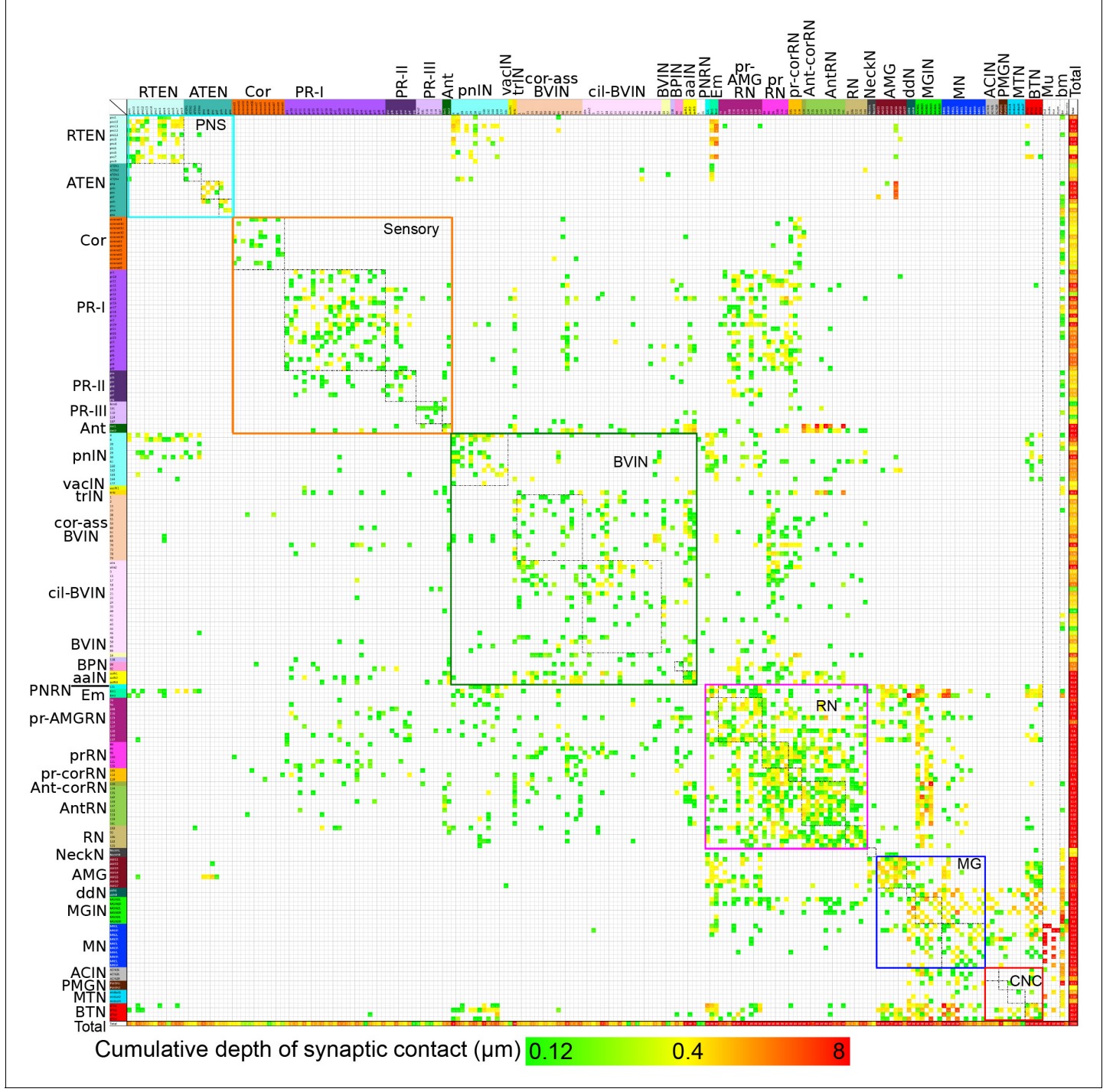

**Figure 16.** Entire connectivity matrix for the complete brain of a larva of *Ciona intestinalis*. Shown for all synapses are the pre- (rows) and post-(columns) synaptic cells, colour-coded by cell type (see *Figure 1—source data 1*) and arranged in their rostro-caudal sequence along the longitudinal axis (presented in *Figure 16—figure supplement 1* is the same matrix with cells sorted into left and right sides). Each intercept is colour-coded for the cumulative depth of presynaptic contacts made by that neuron upon its postsynaptic partner (key, bottom). In the case of dyads or triads, all connections are plotted. Also included are muscle cells, and the basal lamina of the CNS, both of which are exclusively postsynaptic. Other cell types, particularly ependymal cells lacking axons, are excluded. Muscle cells of the dorsal and medial bands are pooled on each side, because these are connected via gap junctions (*Bone, 1992*). The matrix is bounded by nested boxes between specific cell types. The smaller boxes enclosed by dashed lines indicate the connections between neurons of the same subtype. These are enclosed within boxes bounded by coloured lines, which indicate connections between neurons of the same brain region. Neurons of the brain vesicle are segregated into sensory neurons (orange lines, Sensory),

*Figure 16 continued on next page*

*Figure 16 continued*
intrinsic interneurons (pink lines, BVIN) and relay neurons (green lines, RN). Remaining boxes are as follows, neurons of the: PNS (PNS); motor ganglion (MG); and tail neurons of the caudal nerve cord (CNC). For matrix file see (*Figure 16—source data 1*).
The following source data and figure supplements are available for figure 16:

**Source data 1.** Matrix in *Figure 16* 1 as excel file.
**Source data 2.** Matrix in *Figure 16—figure supplement 2* as excel file.
**Figure supplement 1.** Matrix of connections from *Figure 16* sorted by left and right sides.
**Figure supplement 2.** Entire matrix of putative gap junctions for the complete brain of a larva of *Ciona intestinalis*.

neurons' interneurons to motor neurons, both from AMGs in the dorsal MG, and as reported from the bipolar tail neurons (*Stolfi et al., 2015*). It is not yet clear whether the cilia of DCENs or VCENs may alternatively have thermoreceptive, electroreceptive, or chemoreceptive properties, although there is evidence for chemotactic behavior during larval settlement, mediated by epidermal sensory neurons.

A proposed central pattern generator (CPG) for *Ciona* has been compared (*Horie et al., 2010*) to the vertebrate swimming CPG of the lamprey (*Grillner and Wallén, 1999*) but in *Ciona* omits the role of excitatory interneurons in general and ipsilateral connections within the motor network in particular. Moreover the depicted network is left-right symmetrical and with proposed direct contralateral input to motor neurons, features we now show to be lacking in *Ciona*'s CNS connectome. This difference highlights the power of a complete connectome to reveal the actual connections between identified neurons. We find the left-right difference of the ACIN connections, in particular, to be most surprising, and endorsed by the failure of the right ACIN to send a neurite into the left neuropile even though it crosses the midline. Additional features of the network's right side also support this asymmetry: the lack of ACIN1 on the right side, and the two additional right-side interneurons with neurites that remain ipsilateral, and the lack of inhibitory feedback to the first interneuron on the right from the ipsilateral ACIN. An additional left-right asymmetry is provided by the input from the first interneurons MG1 L and R to the second interneurons MG2 L and R, which is far greater on the left than on the right side, and supported by an asymmetrical distribution of gap junctions. The latter is endorsing, and unusual because putative pathways formed by gap junctions are in general more symmetrical than those formed by chemical synapses.

Larvae swim in a helical pattern (*McHenry, 2005*) from simple bilateral flexions of the tail (*Bone, 1992*; *Nishino et al., 2010*). The helical

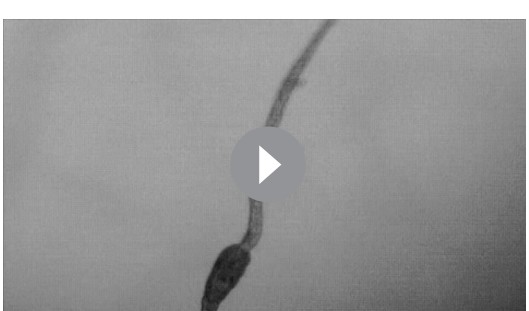

**Video 7.** Unilateral tail flick. A larva exhibits a unilateral tail flick.

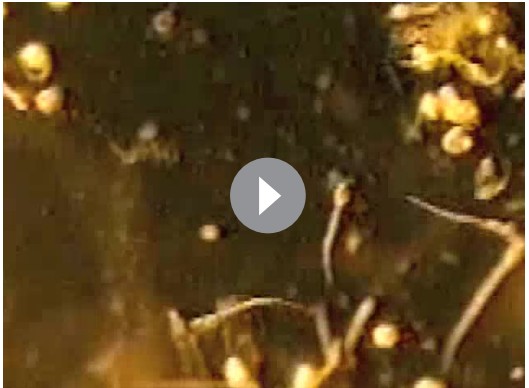

**Video 8.** Asymmetrical tail flicks. A larva exhibits repeated tail flicks to the same side of the trunk.

pattern may reflect the asymmetry of motor pathways, but varying the direction of incident light relative to the ocellus could also generate helical klinotaxis, enablng phototaxis in response to cyclical changes in light intensity as the shadow of the ocellus pigment sweeps across the outer segments (*McHenry and Strother, 2003*; *McHenry, 2005*; *Randel and Jékely, 2016*). Based on gravitaxis in young larvae (*Tsuda et al., 2003*) we also predict that the antenna neurons' circuits underlie a directional response to gravity. It is interesting that the visual pathway originating in the right-sided ocellus and the gravity pathway from the antennal cells, both converge asymmetrically in the motor ganglion, the visual pathway stronger to the left of the ganglion and the gravity pathway to the right. These asymmetries in sensory input to the MG, particularly for the antenna pathway, suggest that a sensory input ought to generate a sided swimming movement, such as a large unilateral contraction (*Video 7*). We predict from the network that this should predominantly be on a fixed side of each animal, as video recordings indeed suggest (*Video 8*; Ryan, unpublished observations).

Despite its shared evolutionary ancestry with chordates, the synaptic network we report incorporates many pathways that are manifestly left-right asymmetrical but that compare to pathways in the vertebrate brain so far reported to be bilaterally symmetrical in form and function. We interpret these asymmetries as an ascidian specialization brought about by the small cell numbers and rapid development of the larval stage, and compatible with the larva's sensorimotor behaviour and helical swimming pattern.

## Materials and methods

### Animals

Adult sea squirts, *Ciona intestinalis* (L.), were collected by Mr. Peter Darnell from Mahone Bay, Nova Scotia. Adults were kept in tanks under constant illumination at the Aquatron facility of Dalhousie University, with flowing sea-water (5–6 L/min) at ~18°C. Adults kept for 1–5 days were removed from the tank and dissected to expose the oviduct and sperm duct. Eggs were collected from the oviduct using a pipette and placed in Petri dishes containing sea-water filtered through a Nalgene 0.2 μm syringe filter. The animals were then washed with sea-water and the sperm duct pierced with a pipette and sperm sucked directly into the pipette and placed in a microcentrifuge tube. Sperm from one adult was added two drops at a time into a Petri dish containing eggs from a different adult and the dish gently swirled to distribute the sperm. Eggs and sperm were left for 15 min for fertilization to occur, then eggs were rinsed several times with filtered sea-water and placed in a Petri dish with filtered sea-water, wrapped in aluminium foil, and placed in an incubator at 18°C.

### Electron microscopy

Dark-reared larvae were removed from the 18°C incubator after 20 hr, and larvae (21 hr post fertilization/2 hr post hatching) were fixed at 4°C for 2 hr in 1% $OsO_4$ in 0.2M $Na_2PO_4$ (phosphate buffer) adjusted to pH 7.2 with HCl. The animals were then post-fixed in 0.2M phosphate buffer containing 2% glutaraldehyde for 1 hr at 4°C. Prior to fixation newly hatched larvae received light from a dissecting microscope fibre optic illuminator only briefly to check that they were swimming, and to pipette them into the fixative. Fixed specimens were dehydrated in an ethanol series (30%, 50%, 70%, for 10 min each, followed by 90% 95%, 100% then propylene oxide for 5 min each). After dehydration, specimens were placed overnight at 23°C in a dish containing propylene oxide and Poly/Bed 812 resin 1:1. Sibling larvae, the products of a single cross, were next transferred to 100% resin for 3 hr and then placed in 100% resin and polymerized at 60°C for 48 hr. Larvae were sectioned and checked for acceptable fixation. Nervous tissue from marine animals presents special problems for good EM fixation (*Cobb and Pentreath, 1978*) and a single larva selected for its clear synaptic vesicle profiles and intact cell membranes was cut in a series at 60, 70 or 100 nm, as reported in Results (*Figure 1A*). All sections were post-stained for 5–6 min in freshly prepared aqueous uranyl acetate followed by 2–3 min in lead citrate. Sections were viewed using an FEI Tecnai 12 electron microscope operated at 80kV and images captured initially using a Kodak Megaview II camera with software (AnalySIS: SIS GmbH, Münster, Germany), or a later Gatan 832 Orius SC1000 CCD camera using Gatan DigitalMicrograph software (Gatan Inc., Pleasanton, CA).

## Imaging

High magnification 3.85 nm per pixel images were collected for the neuropil region of each section. The profile area ranged from five 5 × 5 montages per section in the posterior brain vesicle to single 2 × 2 montages in the tail. Independent lower magnification 13.9 nm per pixel images of the entire CNS and overlying epidermis were collected for every section in the anterior BV, and for every fourth section through the posterior brain vesicle, neck, motor ganglion and anterior tail.

The montages were compiled automatically with the Gatan DigitalMicrograph software or with AnalySIS. Given a limitation of the Gatan software-generated montages, final montages were also manually montaged in Adobe Photoshop. Images were imported into either a high magnification or low magnification series in Reconstruct (*Fiala, 2005*), and sections manually aligned using this software. All profiles in every third section, somata in the low-magnification series and all profiles in the high-magnification series, were then traced completely. Fewer than five neurites were candidate orphans that lacked synapses, and thus did not contribute to the CNS connectome. Skeletonization of neurite projections was accomplished using the function Z-Trace in Reconstruct, which connects the mid-point of each traced profile. In the high magnification series, traces were hidden and all sections then blindly annotated for synapses, putative gap junctions, dense-core vesicles and cilia. After each block of sections was annotated, the traces were then made visible and annotated elements assigned to specific pre- and postsynaptic elements. Blind annotation was duplicated in 100 section blocks by an independent annotator (Ms Carlie Langille). Most annotations of each viewer duplicated existing synaptic contacts seen by the other, but neither viewer annotated a synapse between two partners that did not replicate a synapse formed elsewhere by the same two cells. Of the differences observed, 95% were simply those between the numbers of sections in which a synapse was observed.

## Synapse analysis

Each neuron was classified from structural criteria, mostly from the identity of its presynaptic partner(s) (see key in *Figure 1—source data 1*). Synapses were identified based on the criteria established in *C. elegans* (*White et al., 1986*) of a cluster of vesicles at a presynaptic membrane. Although postsynaptic densities were observed at some synapses (*Peters and Palay, 1996*), these were either not clear or not present at all sites with a presynaptic vesicle cluster. Putative gap junctions were annotated at sites with juxtaposed membranes and densities on the membranes of both sides, except where such contacts were directly adjacent to the neural canal, which are candidate desmosomes and were not studied further. The numbers of synapses and the numbers of sections in which a synaptic profile, a single imaged cross-section of a synapse, was observed from each were both measures used to quantify synaptic strength, as was their product, the total depth of synaptic profiles. The cumulative depth of synaptic contact was calculated by multiplying the number of sections in which a synapse was observed by the section thickness. These values for each presynaptic neuron were linearly proportional (*Figure 3B*); the relationship remained unchanged for the numbers of synapses per neuron when small synapses – those containing only a single section – were removed. This depth in μm was used for network diagrams as a putative proxy for synaptic strength.

Nucleus location was used to determine neuron sidedness. A line through the midline of the CNS drawn from the ventral to dorsal surfaces through the neural canal was used to classify nuclei as left or right. Those with nuclei intersected by the midline were classified as midline neurons. Nuclear x/y positions are reported in *Figure 1—source data 1* Partitioning the parts of each neuron (cell body/soma, axon, terminal, or dendrite) was determined by examining the reconstructed neuron and identifying the sections at the boundaries between these parts. Axons were distinguished from their somata of origin by the reduction in profile diameter, and terminals were distinguished from axons where we observed branching or expansions with presynaptic sites near their final termination. Many terminals had but small swellings, and their synapses occurred onto collateral terminals or axons, so were considered en passant. Dendrites were those regions proximal to the cell body in which short neurites extended, with or without branching. Some relay neurons contain expanded branched regions near their axon hillock, which we classified as 'BVterm' and these were included in the analysis of dendritic synapse. Exclusively postsynaptic dendrites were lacking in the CNS. Despite our attempts to partition them in this way, neurons were characterized by a lack of

segregation of pre- and postsynaptic sites along their length and a general absence of defining features for each region.

## Video microscopy

For *Video 1* and *Video 7*, larval swimming was recorded using a Motionscope CCD camera by Redlake Imaging (Model PCI 2000 S #1108–0009) mounted on a Leica MZFLIII dissecting microscope equipped with a Plan Apo 1/0x objective and recording these on Sterling computer using MIDAS software. Clips were selected and converted to. mov format using iMovie. For *Video 8*, hatched larvae were recorded using a Leica MZFLIII dissecting microscope equipped with a Plan Apo 1/0x objective, capturing their images with a high resolution CCD camera (Elmo TSM 41OH) on VHS video tapes, and then transforming these to digitized image sequences at up to 60 fps by software (QuickTime Pro: Apple Inc.).

## Acknowledgements

This work has been supported by grant DIS-0000065 (to IAM) from the Natural Sciences and Engineering Council of Canada. We thank Ms Carlie Langille for assistance in proofreading neurite profiles and annotating synapses, Drs. Dianne Nicol (Tasmania), Janice Imai (and Ayami Matsushima (Fukuoka) for their video images of larval swimming, and Ms Jane Anne Horne for assistance with computer techniques. We also acknowledge Dr. Scott W Emmons (Albert Einstein College of Medicine, New York) for help in comparing our findings with the *C. elegans* connectome.

## Additional information

### Funding

| Funder | Grant reference number | Author |
|---|---|---|
| Natural Sciences and Engineering Research Council of Canada | DIS0000065 | Kerrianne Ryan<br>Zhiyuan Lu<br>Ian A Meinertzhagen |

The funders had no role in study design, data collection and interpretation, or the decision to submit the work for publication.

### Author contributions

KR, Conception and design, Acquisition of data, Analysis and interpretation of data, Drafting or revising the article; ZL, Cut the entire ultrathin section series upon which this study is based and provided essential expertise on serial section electron microscopy, Acquisition of data; IAM, Conception and design, Drafting or revising the article

### Author ORCIDs

Ian A Meinertzhagen, http://orcid.org/0000-0002-6578-4526

### Ethics

Animal experimentation: This study was approved by Dalhousie University protocol I9-015.

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
