## [Decision Letter]

Thank you for submitting your article "Sidedness in the brain of a chordate sibling, the tadpole larva of *Ciona intestinalis*" for consideration by *eLife*. Your article has been reviewed by three peer reviewers, and the evaluation has been overseen by a Reviewing Editor and Eve Marder as the Senior Editor. The following individuals involved in review of your submission have agreed to reveal their identity: Jason Pipkin (Reviewer #1); Scott Emmons (Reviewer #2).

The reviewers have discussed the reviews with one another and the Reviewing Editor has drafted this decision to help you prepare a revised submission.

Summary:

Your manuscript provides an important report of the connectome of the larva of *Ciona intestinalis*, a chordate with a very small number of neurons. You present the result of a tremendous effort reconstructing the connectome of larval Ciona intestinalis from a series of electron micrographs. This represents the second whole-organism connectome ever fully reconstructed and is a finding of major significance.

The reviewers were universally impressed with the potential significance of this work, and were enthusiastic about is eventual publication. Nonetheless, because of its unusual and rare nature, and therefore its potential value to the field, the reviewers were very concerned that the data be presented as clearly and transparently as possible and that the manuscript itself be brought to the same high quality as the work itself. We all understand the inherent difficulty of figuring out how to present this amount of data in an accessible form, and we also understand that after a project that must have taken years of work, that you must be ready to see it in print. But all three of the reviewers found the presentation less than ideal. I am loath to tell you exactly how to reorganize it. I am taking the unusual step for *eLife* and including the full reviews so that you can both appreciate the enthusiasm of the reviewers and also see first-hand the myriad issues that arose during the discussion and the reviews. I understand that addressing some of the issues raised below may lengthen the paper. That is ok, as long as you use appropriate headings to orient the reader and to provide signposts to the work. Obviously, the ideal would be to achieve greater transparency and clarity without too much additional length.

Most importantly, please read and think about all of these comments, and then do your best to prepare a revision that deals with these issues. You will note that some issues arise in all three reviews, such as a clearer description of the system and its behavior. Please pay especial attention to those issues that are brought up in multiple reviews, as they are crucial to address. Many of the comments can be addressed with a few words, while others will require more thought.

Important: *eLife* would like you to release in an appropriate location the source data and reconstruction data and identifying criteria for use by the community. The major image dataset and reconstructions together seem an ideal candidate for sharing via the Open Connectome Project. Ideally, as much of the original data and the reconstructions should be openly available so that the connectome becomes a working data set for whomever is interested. Please let us know what your plans are in this regard.

*Reviewer #1:*

Ryan et al. present the result of a tremendous effort reconstructing the connectome of larval *Ciona intestinalis* from a series of electron micrographs. This represents the second whole-organism connectome ever fully reconstructed and is a finding of major significance. The authors relate their findings within the context of structural and connectivity asymmetries, particularly emphasizing relevant comparisons to vertebrates. As many readers are unlikely to be familiar with *C. intestinalis*, the Introduction should discuss the life cycle of the larva and its behavior. My understanding is that younger larvae swim upwards and older larvae swim downwards to eventually settle based in part on sensors for gravity and light. The authors should emphasize throughout how the *Ciona* connectome relates to its behaviors.

The generalizability of the conclusions drawn about the asymmetry here are tempered by the single-organism sample size. There is space for even more discussion about how variable these left-right asymmetries are from animal-to-animal. What percentage of larvae eventually settle? Perhaps the larval connectome can be "noisy" in terms of cell number and connection strength if only a small number of larvae need to survive and eventually become adults.

Some relevant comparisons to other connectomics work are missing. For instance, there are several recent papers from Cardona's group on larval *Drosophila* that briefly touch on issues of right-left symmetry (sometimes only in supplemental material). Their work, along with Randel et al. (2015), rely on an assumption of symmetry to determine the confidence that a given synaptic connection is real. This work challenges that assumption, and it would be good to address this contrast.

The manuscript as currently written lacks adequate elaboration of its methods. Neurons are divided into types, soma position is defined as left, right, or medial, and neuronal arbors are divided into different parts (axon, terminal, etc.) without any explicit and rigorous description of the criteria by which these decisions were made.

The authors report a measure of synaptic strength that depends on the number of sections containing a presynaptic profile. However, section thickness varied and it is not clear that the necessary compensations were done. Inevitably, all anatomically-based hypotheses of synaptic strength fall short of direct physiological measurement, so the authors must make absolutely clear at all times what exactly is being measured. The authors describe synapses "onto" basal lamina, a non-cellular structure. Because the concept of a synapse is ingrained the minds of many readers as involving at least two cells (excepting autapses), these should be described differently, perhaps as "putative sites of neurotransmitter release." Could these be sites of neuromodulator release?

There are at least two discrepancies in the presented results. First, there are missing connections in Figure 12 that are present in 12A. Second, the number of polyadic synapses is variously reported as 920, 921, and 2150 (25% of 8601) in the Abstract, Results, and Table 1 respectively.

Finally, the manuscript would greatly benefit from a key that relates neuron type and abbreviation.

*Reviewer #2:*

Ryan, Lu, and Meinertzhagen present the complete connectome of the larval ascidian *Ciona intestinalis*. This is a paper of potentially historic importance. Not only is this the second whole-animal connectome deciphered, it is the first connectome to be obtained from a single individual. Moreover, Ciona is a chordate, having a notochord and dorsal nerve cord. Therefore, the features of the connectome are of particular interest as a point of comparison to the previously available connectome of *C. elegans*. When there was only a single connectome available, and that of a tiny, potentially highly-derived invertebrate, it was difficult to know what to make of it and whether one could generalize its characteristics to other animals. Even if *C. elegans* were representative of invertebrates, how similar would the nervous system of an animal from the other great branch of metazoans be?

In fact, interestingly, Ryan et al. find they are very similar! Similar features include similar overall cell numbers but a rather large number of cell types, largely unbranched neurons making en passant synapses, similar numbers of chemical and gap junction synapses, many polyadic synapses, and little segregation of neurons into separated axonal and dendritic regions, indicating graded potential neurons. In the beginning of the Discussion the authors claim that many features of the connectome are basal. These may indeed be, but there really is little support for this claim. As there are no other connectomes to compare to apart from *C. elegans*, how can we know, for example, that "the apparent general lack of precision" is basal? The only support mentioned for one feature being basal is that unpolarized synapses are seen in coelenterates. But they are also seen in a pulmonate mollusc. Is the pulmomate mollusc primitive or basal?

Summary of substantive concerns

An important feature of the *C. elegans* connectome is that it comprises a neural network making a single giant component. That is, everything is connected to everything, eventually. One gathers this is true for the *Ciona* connectome as well (Discussion, first paragraph: features include the "wide range and combination of cells that share synaptic contacts"), but it is not discussed. In general, the authors should pay more attention to the overall graph properties of the network. The degree distributions are particularly important and should be given.

The authors have chosen to focus their manuscript on left/right asymmetries in the connectome. This is an over-emphasis at the expense of the other interesting features in the opinion of this reviewer. The Introduction does not tell us why this is such an important issue. It begins by suggesting such asymmetries are commonly known. Then later on (last paragraph) it is said they are rare and difficult to recognize. Apparently the issue of l/r asymmetry has been well-studied in ascidians and results from asymmetric nodal expression during embryogenesis as it is in vertebrates. Most of the Discussion is devoted to this issue. Since we know many morphological features of the body are asymmetric, we can expect this will be true for the nervous system as well; it's not clear why so much is made of this here.

Representing as it does a descriptive presentation of a very large number of observations, it is important for the manuscript to give the reader a framework that is easy to grasp and keep in mind into which the more detailed observations may be placed. This is particularly so in this instance, as very few readers will have any prior knowledge of the *Ciona* larva. Figure 1 is not clear and sufficient in this regard. It should have an excellent diagram of the entire larva, showing the locations of sensory structures, ganglia, nerve tracts, and muscles. Since the neurons are unbranched, do they run in bundles with a constant set of neighbors as they do in *C. elegans*? There should be a description of the larva's behavior. One finds out in the Discussion (fifth paragraph) that it does not feed! That's going to make a big difference in the complexity of the nervous system. What does it do besides swim? Does it chemotax, thermotax, and so forth? What sensory-behavioral pathways are revealed in the connectome? What circuits governing swimming behavior and its control?

Here are some additional questions or comments that occurred to this reviewer:

1) What constitutes the CNS and the PNS? CNS is referred to early in the paper without definition. Figure 15 shows that there are also a small number of PNS neurons. How are these distinguished in such a small nervous system? There is an error in the second paragraph of the Discussion, where it is stated there are 302 neurons in "most of the CNS" of *C. elegans*. There are 302 neurons in the entire nervous system and there is no clear division in *C. elegans* between CNS and PNS; these terms are not used.

2) Neuron types: this concept is used throughout, but how neuron "type" is defined is not described. Do the neurons really fall into a set of discreet, unambiguous types? In *C. elegans*, the broad classifications sensory neuron, interneuron, motorneuron are not unambiguous. Many neurons generally considered to be interneurons because of the large number of synapses they make with other neurons also make nmj's, while many neurons considered to be motor neurons also have lots of synapses onto other neurons. "Motor neurons" can have stretch receptors, and "sensory neurons" can have synapses onto muscle. Careful examination reveals that where the cut-offs should in fact be among these broadest classes is unclear and may be somewhat arbitrary. When the manuscript first refers to different neuron types (subsection “Neurons”: brain vesicle intrinsic interneurons, MG interneurons, BV relay interneurons), the reader does not know what these neuron types are. So the descriptions of their various properties are hard to appreciate. Diagrams of the various types, what defines them, where their cell bodies lie, and where their processes go, are needed.

3) In the first paragraph of the subsection “Neurons”, how can a "terminal" make an en passant synapse?

4) In the third paragraph of the subsection “Synapses”; junctions <2 sections were omitted from consideration. What is the justification for this cutoff? Why not junctions <3? How many junctions does this exclude and what fraction of the "load" do they carry?

5) In the fourth paragraph of the subsection “Synapses”, and throughout the paper: average is used as a statistic (average 37 presynaptic sites per CNS neuron). This doesn't seem to be a meaningful statistic. The distributions need to be given. In this regard, Figure 3 should probably be a semi-log survival curve. The curve shown doesn't tell us much as most of the points either don't move much along the X-axis or much along the Y axis.

6) Figure 3 is described in the text (subsection “Synapses”, fifth paragraph) as showing the number of presynaptic sites versus the number of postsynaptic partners, but the axes in the figure are labeled number of synapse profiles vs number of synapses. What are synapse profiles? And why is this ratio of interest? What's the point being made here, that all neurons have the same density of synapses?

7) In the second paragraph of the subsection “Sidedness in the brain vesicle”: there are said to be 17 enigmatic coronet cells on the left side, and Figure 5 is referred to. But I cannot see 17 enigmatic coronet cells in this figure. In fact, it's hard to distinguish any cells in the figures produced by Reconstruct (I presume). There need to be drawn diagrams to accompany them. And the drawings should be shown in the context of the entire larva, like the drawings of *C. elegans* neurons in WormAtlas (e.g. http://wormatlas.org/neurons/Individual%20Neurons/AVAframeset.html).

8) Figure 12. Can't read the labels in A and B. How do we know these left/right differences aren't just developmental noise?

9)In the sixth paragraph of the Discussion: number of neuron types. This seems to me to be Results, not Discussion.

10) I can't relate the cells listed (or cell types?) in the legend to Figure 7 to the cell types (?) listed in Table 2. I can't relate the cells listed in the master cell list Figure 3—figure supplement 1 with the cells e.g. in Figure 3. The master cell list Figure 3—figure supplement 1 is said to be an excel file, but mine's a PDF. Excel file would be much easier to work with (e.g. the heading row can be kept visible as one scrolls down the list).

11) I don't have a Figure 9—figure supplement 1.

12) Table 1 lists 3 polyadic gap junctions, but polyads are not defined for gap junctions because there is no defined pre- and post-synaptic cell.

13) Figure 15. This is the main presentation, along with individual neuron maps (missing), of the findings of this reconstruction. This is presumably the directed graph of chemical connections. The corresponding undirected graph of the gap junction connections needs to be included. I don't understand why the weights of the matrix are given as "percentile synaptic strength." In fact, I don't know what this means, rank order percentile? or percent of the highest value? Is a matrix with numbers (number of EM sections, or physical size in microns, since section thickness varied across this reconstruction) available rather than these colors? Modelers or theoreticians who would like to study the properties of the network (there will be many!) will need this.

*Reviewer #3:*

This is a beautiful and very important work describing the entire neuronal connectome of the *Ciona* tadpole larva. *Ciona* early development and neuronal development are very well understood and the full larval connectome is a milestone addition to the resources available for this important chordate model organism. Importantly, urochordates are the sister group to the vertebrates, and have been studied intensively to understand the evolution of key vertebrate features. The connectome also provides a very important reference point for broader circuit comparisons across animals.

Given that this is a full connectome paper, I think the authors should not feel constrained by the 5000 word limit, but aim to describe the neuron complement and the full circuits in more detail. They should also think about how they can relate the connectivity to known larval behaviors (e.g., phototaxis, gravitaxis, larval settlement).

First, larval behaviors:

There is one important mistake that the authors need to correct first to be able to better interpret the connectome.

They state in the paper twice that the tadpole swims in a bilateral fashion with tail undulations: "The apparent symmetry of the characteristic swimming pattern of chordate-like tail undulations" and "Larvae exhibit a simple bilaterally symmetrical swimming response to light "

This is not true, *Ciona* and other ascidian larvae swim in a helical pattern. This was already known to Eakin and Kuda (1971) who wrote "When an ascidian larva is actively swimming the body rotates about its long axis." More importantly, a detailed kinematic analysis of swimming of the *Ciona* tadpole revealed how the larva turns towards light during helical swimming.

The authors should consult and cite "The kinematics of phototaxis in larvae of the ascidian Aplidium constellatum" by McHenry and Strother (Marine Biology, 2003) and "The morphology, behavior, and biomechanics of swimming in ascidian larvae" by McHenry (Can. J. Zool. 83: 62-74 2005). Appreciating the correct motor pattern is important to correctly interpret how the circuits work. The helical pattern emerges due to the asymmetry in tail bending. This is likely one of the main reasons for the high degree of asymmetry in this animal.

Video 1 does not demonstrate bilateral swimming, it is actually impossible to tell from this video what the locomotor pattern is (bilateral or helical) because the larva moves out of focus and the frame rate is too low. This video may be removed or just used as an illustration.

Neuron types and connectivity:

Figure 3—figure supplement 1 lists all cells and their important features. Since it is a comprehensive paper, the morphology and position of at least one example from each cell type should be shown in a supplementary figure (or figures) and where each cell type is located in the body (show body outline).

Likewise, some of the anatomical diagrams are difficult to interpret because the cells are not shown in the context of the whole body or together with other cells from the same circuit.

Same for the videos: Video 2 and Video 3 – the detail of the ocellus should be embedded into the larger anatomical context (e.g., by showing the body outline).

It is in general difficult to follow the description of connectivity of the different subsystems (e.g., eyes, otolith), because the relevant details are embedded in the summery diagrams and are not explained in full. I would really encourage the authors to describe the different sensory-motor systems in more detail. This should include the reconstructed neurons in the whole-body context (anatomical diagrams) and the corresponding circuit diagrams. For example, what is the full cell complement and wiring diagram of the PRC circuit? Show type I, II, III PRCs and their downstream circuitry, to the muscles. Likewise, show the otolith and coronet circuits. What are the PNS neurons and their circuitry? How about the tail circuitry that probably tunes the left-right motor pattern?

The integration can then be described following the details of the subsystems (as is now e.g., in Figure 9).

It would also be very interesting to see the strength of all reciprocal connections (e.g., represented as the geometric mean of 2-way synapse numbers) at the level of the entire connectome. This can easily be computed from the whole matrix.

It is hard to read Figure 7—figure supplement 1. The network should be shown in higher magnification. An alternative representation that would be useful would be to merge cells of the same type and connectivity into one node and show the average synaptic strength.

Figure 11. Also show cell morphology of MN classes. Please also show a schematic of the different muscle types and their position relative to the MN axons. It is not clear how an MN projecting on the left side can synapse on muscles both on the left and right sides of the body.

Figure 15 is hard to read – please label neuron categories also on the presynaptic side. Provide list of abbreviations. Label major categories in the figure (corresponding to the boxes e.g., sensory neurons).

Discussion:

The functional and comparative implications of the *Ciona* connectome should be spelled out in more detail. For example, how does the connectome explain negative phototactic behavior? The strategy of ascidian phototaxis is helical (not visual, for an explanation see Randel Philos Trans R Soc Lond B Biol Sci. 2016 doi: 10.1098/rstb.2015.0042.). It is known that upon light off, the tail begins to vibrate (Grave, 1921; Dilly, 1964). This would mean that when the helically swimming tadpole ocellus turns away from the light source, it initiates tail movement. What can one deduce from the connectome? How does it compare to other phototactic systems (fish, lamprey, annelid)?

The authors interpret the one sidedness of the eye as a reduction in one side, which is likely correct, and may have been possible as the larva lost visual phototaxis (which requires two eyes) and evolved helical phototaxis.

Gravitaxis has also been studied by otolith ablations. Is the motor pattern known? Does the connectome explain how gravitaxis works?

[Editors' note: further revisions were requested prior to acceptance, as described below.]

Thank you for submitting your article "Sidedness in the brain of a chordate sibling, the tadpole larva of *Ciona intestinalis*" for consideration by *eLife*. Your article has been reviewed by three peer reviewers, and the evaluation has been overseen by a Reviewing Editor and Eve Marder as the Senior Editor.

The reviewers have discussed the reviews with one another and the Reviewing Editor has drafted this decision to help you prepare a revised submission.

The major issue is that all three reviewers feel that this paper has the potential to be a classic, and therefore they have strong feelings about further improving it for the benefit of readers for years to come. Everyone involved in the review understands that the wealth of information in the raw data makes deciding how to organize the paper difficult. That said, we believe that one last round of revision will benefit the paper and its utility to future readers. We fully appreciate that this paper is yours, and that honest differences of opinions might arise between well-intentioned reviewers and authors. That said, the reviewers are reading the paper without benefit of all of the authors' working knowledge of the data, and therefore, their input provides crucial information of how other readers may confront the paper.

*Reviewer #2:*

This paper is improved after the first revision. However, it remains difficult for the general reader to follow. The difficulty lies in the fact that it is basically descriptive with many different neuron types referred to.

For the general reader, there needs to be a more careful and complete description of the work. First of all, the paper does not present an "entire synaptic connectome" (Abstract) or "full synaptic connectome" (first sentence of Discussion). It presents only a partial connectome, that of the CNS. This is just the beginning of how the presentation gets the reader off the track. Subsequently, CNS, PNS (also described as "epidermal"), and non-neuronal ependymal and support cells are all mixed in together in the descriptions. While connectivity, network properties and structure, and neuronal pathways are described, much more of the manuscript is devoted to individual neuron descriptions. Indeed, in the subsection “Sidedness in CNS pathways”, referring to some of their pathway diagrams, they state "However, these shortest paths fail to depict the complexity of integration revealed in the total network." The real emphasis can be seen in the subsection “Asymmetry in cellular composition”, where it out with the sentence "The overall cell complement is closely similar on the two sides (left: 125; right: 129; midline 46)." Total: 300, not 177 as we read in the Abstract. It is really a paper on the cellular structure of the *Ciona* larval nervous system, with emphasis on its left-right asymmetry, as properly presented in the title. Most of the Discussion is devoted to the developmental origin of the asymmetry.

For what they're worth, I offer the following further comments and suggestions for the authors' consideration.

1) I still don't understand the significance of the issue of left-right asymmetry. In the first paragraph of the Introduction, after the sentence that ends "and so provide a useful model to study many aspects of brain asymmetry," I need a next sentence that begins: "This issue is important because…" In the Abstract: "Chordate in body plan and development, the larva provides by contrast an outstanding example of brain asymmetry." By contrast to what?

2) The sentences that begin the Results section belong in Methods. Similarly, the second paragraph of the subsection “Synapses”. Results need to begin: "The nervous system of the *Ciona* larva consists of.…" We need a general description of the larva and its nervous system so that when the EM series (and everything else) are described we know where we are. For example, for the general reader, "starting at the level of the otolith pigment.…" is meaningless as we don't know where that is. The major nervous system partitions given in the first paragraph ("posterior motor ganglion," "anterior brain vesicle," "motor ganglion," need to be shown in Figure 1.

3) What does it mean in the subsection “Synapses”, that "A total of 301 cells of the CNS were imaged,"? Wasn't every cell in an EM cross section "imaged"? Do you mean reconstructed or traced? The issue of the ependymal cells should not be taken so lightly. Indeed, the authors feel the need to define these as "those ciliated cells abutting the canal that lack an axon," but they only do so in passing in the subsection “Asymmetry in cellular composition”. And apparently this work shows some cells previously thought to be ependymal are in fact neuronal. In the subsection “Synapses” it is explained that there are at least 30 CNS cells that were not studied. As we don't know whether these are neurons, ependymal, or other support cells, we don't really know that "the entire CNS included 177 neurons". Maybe it includes 207.

4) Relay neurons. In such a highly cross-connected connectome as this can the authors really distinguish "relay" interneurons from other types? Figure 11 labels some neurons "interneurons" and some "relay" neurons. This distinction needs to be discussed.

5) Explain and emphasize at the top that PNS neurons are often referred to as "epidermal."

6) In the second paragraph of the Discussion: delete "most of".

7) The authors say they have determined the connectivity of 177 neurons. But their table of network statistics, Table 2, says there are 213 nodes in the chemical network and 193 in the gap junction network. Please explain this discrepancy.

---

## [Author Response]

[…]

*Reviewer #1:*

*Ryan et al. present the result of a tremendous effort reconstructing the connectome of larval Ciona intestinalis from a series of electron micrographs. This represents the second whole-organism connectome ever fully reconstructed and is a finding of major significance. The authors relate their findings within the context of structural and connectivity asymmetries, particularly emphasizing relevant comparisons to vertebrates. As many readers are unlikely to be familiar with C. intestinalis, the Introduction should discuss the life cycle of the larva and its behavior. My understanding is that younger larvae swim upwards and older larvae swim downwards to eventually settle based in part on sensors for gravity and light. The authors should emphasize throughout how the Ciona connectome relates to its behaviors.*

We now added the following text to the Introduction:

“*Ciona* releases 5000-10000 eggs per individual (Peterson and Svane, 1995) and its eggs are released either individually, or in a mucous string (Svane and Havenhand, 1993). […] Of the reported behaviors, the shadow response, in which a dimming of light results in symmetrical swimming, is the best studied, developing at 1.5 hph and increasing in frequency after 2 hph (Zega et al., 2006).

In addition, we address the issue of how the *Ciona* connectome relates to its behaviors, later, in the Discussion.

*The generalizability of the conclusions drawn about the asymmetry here are tempered by the single-organism sample size. There is space for even more discussion about how variable these left-right asymmetries are from animal-to-animal. What percentage of larvae eventually settle? Perhaps the larval connectome can be "noisy" in terms of cell number and connection strength if only a small number of larvae need to survive and eventually become adults.*

Of course we would love to address the issue of the constancy of the connectome in sibling larvae, and we address these rather general points on how generalizable our conclusions are with the following minor additions:

“Additionally, we cannot exclude the possibility that minor left/right asymmetries might be the product of developmental noise, insofar as only a small number of larvae need to survive and eventually become adults.” (Discussion) and:

“Our conclusions are based on a single larva, although four sibling larvae have been reported with closely similar overall cell complements (Nicol and Meinertzhgen, 1991). Data are still lacking on the constancy of neuron cell types and their connections between individuals. The consistency of asymmetrical differences will only be resolved when additional sibling larvae are examined, as we now undertake.” (Discussion).

*Some relevant comparisons to other connectomics work are missing. For instance, there are several recent papers from Cardona's group on larval Drosophila that briefly touch on issues of right-left symmetry (sometimes only in supplemental material). Their work, along with Randel et al. (2015), rely on an assumption of symmetry to determine the confidence that a given synaptic connection is real. This work challenges that assumption, and it would be good to address this contrast.*

We now added the following text:

“Some connectomic analyses rely on symmetry of the connections between left and right sides to validate the synaptic connections of neurons that are paired (Durbin, 1987; Randel et al., 2014; Ohyama et al., 2015). This approach has not been possible in *Ciona*, because many cells are not bilaterally paired in the brain vesicle, while in the motor ganglion where cells are paired, presynaptic inputs from the brain vesicle are similarly asymmetrical.”

*The manuscript as currently written lacks adequate elaboration of its methods. Neurons are divided into types, soma position is defined as left, right, or medial, and neuronal arbors are divided into different parts (axon, terminal, etc.) without any explicit and rigorous description of the criteria by which these decisions were made.*

We now added the following text:

*“*Nucleus location was used to determine neuron sidedness. A line through the midline of the CNS drawn from the ventral to dorsal surfaces through the neural canal was used to classify nuclei as left or right. […] Despite our attempts to partition them in this way, neurons were characterized by a lack of segregation of pre- and postsynaptic sites along their length and a general absence of defining features for each region.”

*The authors report a measure of synaptic strength that depends on the number of sections containing a presynaptic profile. However, section thickness varied and it is not clear that the necessary compensations were done.*

We are grateful for this comment, and compiled depth determinations for each synapse were made as suggested, and we have now added these in the text. As it turns out, the corrections made little difference to the overall analysis. To clarify the issue, however, we added the following text:

*“*The depth of synaptic contact was calculated by multiplying the number of sections in which a synapse was observed by the section thickness. […] This depth in µm was used for network diagrams as a proxy for synaptic strength.”

We also changed the chart and R^2^ value in the legend to Figure 3.

*Inevitably, all anatomically-based hypotheses of synaptic strength fall short of direct physiological measurement, so the authors must make absolutely clear at all times what exactly is being measured. The authors describe synapses "onto" basal lamina, a non-cellular structure. Because the concept of a synapse is ingrained the minds of many readers as involving at least two cells (excepting autapses), these should be described differently, perhaps as "putative sites of neurotransmitter release." Could these be sites of neuromodulator release?*

We acknowledge this issue, and exclude these sites of synaptic vesicle release from network analysis, and we provide comparisons of summary data with and without “sites of putatative release” onto the basal lamina in Table 1 (see below). On the other hand, synaptic sites formed onto the basement membrane, and thus without obvious postsynaptic partners, have a presynaptic ultrastructure that resembles other synapses in the CNS, and also resemble neuromuscular junctions where a basal lamina ensheathes the muscle cell that neurotransmitter must cross to act postsynaptically. Other cases where synapses form onto a basal lamina may lack an adjacent muscle cell, and indeed may be sites of neuromodulator release. This possibility seems particularly likely in the case of coronet cells, which express reporters for dopamine and which have exclusively dense-core vesicles at their “synapses”.

We therefore now add the following text:

“Synaptic sites that occur onto the basal lamina, and that therefore lack postsynaptic partners, resemble other presynaptic sites in the CNS, and resemble neuromuscular junctions, where neurotransmitter must cross the basal lamina to act on adjacent muscle cells. […] This is particularly likely in the case of coronet cells, which express reporters for dopamine and have exclusively dense-core vesicles at their synapses onto the basal lamina.”

*There are at least two discrepancies in the presented results. First, there are missing connections in Figure 12 that are present in 12A. Second, the number of polyadic synapses is variously reported as 920, 921, and 2150 (25% of 8601) in the Abstract, Results, and Table 1 respectively.*

These results were based on different subsets of neurons, and have been corrected in the text and tables, and the following comparative data have been added as Table 1. Subsumed by data in the Table, the numbers of synapses are now as follows:

Of ALL synapses recorded: Total 8617;% Reciprocal 5.2%;% Polyad 10.7%

Of ALL without bm, Ep, Mu, space: Total 6661; Reciprocal 6.7%; Polyad 13.8%

Of synapses >1 section without bm, Ep, space:

Without Mu: Total 4913/1124.4; Reciprocal 6.7%/7.8%; Polyad 14%/12.3%

With Mu Number of synapses: Total 6083; Reciprocal 330 (5.4%); Polyad 692 (11.4%); Mu 1170 (19.2%)

With Mu Depth of total synapses: Total 1559.7; Reciprocal 88.28 (5.7%); Polyad 138.8 (8.9%); Mu 435.364 (27.9%)

*Finally, the manuscript would greatly benefit from a key that relates neuron type and abbreviation.*

We see now that the key requested by the reviewer was in fact not appended to the original submission, and is now attached as a supplementary summary data table (Figure 1—figure supplement 2). This was our mistake. An additional, enlarged version of this key with annotated cartoons of neuron shape will be deposited in Open Connectome.

*Reviewer #2:*

*Ryan, Lu, and Meinertzhagen present the complete connectome of the larval ascidian Ciona intestinalis. This is a paper of potentially historic importance. Not only is this the second whole-animal connectome deciphered, it is the first connectome to be obtained from a single individual. Moreover, Ciona is a chordate, having a notochord and dorsal nerve cord. Therefore, the features of the connectome are of particular interest as a point of comparison to the previously available connectome of C. elegans. When there was only a single connectome available, and that of a tiny, potentially highly-derived invertebrate, it was difficult to know what to make of it and whether one could generalize its characteristics to other animals. Even if C. elegans were representative of invertebrates, how similar would the nervous system of an animal from the other great branch of metazoans be?*

*In fact, interestingly, Ryan et al. find they are very similar! Similar features include similar overall cell numbers but a rather large number of cell types, largely unbranched neurons making en passant synapses, similar numbers of chemical and gap junction synapses, many polyadic synapses, and little segregation of neurons into separated axonal and dendritic regions, indicating graded potential neurons. In the beginning of the Discussion the authors claim that many features of the connectome are basal. These may indeed be, but there really is little support for this claim. As there are no other connectomes to compare to apart from C. elegans, how can we know, for example, that "the apparent general lack of precision" is basal? The only support mentioned for one feature being basal is that unpolarized synapses are seen in coelenterates. But they are also seen in a pulmonate mollusc. Is the pulmomate mollusc primitive or basal?*

This has been an object of discussion between the authors and internal reviewers and on balance we are persuaded to remove the term ‘basal’ from the text, as being indefensible. We really mean ancestral, but all features of known nervous systems derive from extant species that don’t allow observation of ancestral forms. We have simply removed the word “basal” or, in the Abstract, and substituted “lack structural specialization(s)” in the text.See also the same point raised by Reviewers 1 and 3.

*Summary of substantive concerns*

*An important feature of the C. elegans connectome is that it comprises a neural network making a single giant component. That is, everything is connected to everything, eventually. One gathers this is true for the Ciona connectome as well (Discussion, first paragraph: features include the "wide range and combination of cells that share synaptic contacts"), but it is not discussed. In general, the authors should pay more attention to the overall graph properties of the network. The degree distributions are particularly important and should be given.*

We have added the following text:

“The network forms a single connected component, with all cells (nodes) being connected by a synapse (edge) to another node in the network (Figure 3—figure supplement 2). The network statistics (Table 2) reveal that the characteristic path length between two neurons is 2.7 (from one neuron, through one other to its target), with neurons having an average of 20 neighbors (synaptic partners) and an overall average network clustering coefficient (existing edges between neighbors of a neuron/possible edges between neighbors of a neuron) of 0.333.”

*The authors have chosen to focus their manuscript on left/right asymmetries in the connectome. This is an over-emphasis at the expense of the other interesting features in the opinion of this reviewer. The Introduction does not tell us why this is such an important issue. It begins by suggesting such asymmetries are commonly known. Then later on (last paragraph) it is said they are rare and difficult to recognize. Apparently the issue of l/r asymmetry has been well-studied in ascidians and results from asymmetric nodal expression during embryogenesis as it is in vertebrates. Most of the Discussion is devoted to this issue. Since we know many morphological features of the body are asymmetric, we can expect this will be true for the nervous system as well; it's not clear why so much is made of this here.*

We chose left/right asymmetries as the most distinctive feature of ascidian larval brains, of greatest interest to a general readership, and think it represents one outcome of reducing cell number, allowing each side of the CNS to specialize in a particular function, especially a sensory function, and for the opposite side to lack the cells for that function. But this perspective is, we think, too speculative and insufficiently warranted to present in a concise revision of our manuscript.

*Representing as it does a descriptive presentation of a very large number of observations, it is important for the manuscript to give the reader a framework that is easy to grasp and keep in mind into which the more detailed observations may be placed. This is particularly so in this instance, as very few readers will have any prior knowledge of the Ciona larva. Figure 1 is not clear and sufficient in this regard. It should have an excellent diagram of the entire larva, showing the locations of sensory structures, ganglia, nerve tracts, and muscles.*

We agree, and now present a new version of the entire larva, with most colour-coded cell types to which the account refers, in a new panel (Figure 1). Reference to existing panels has been revised in the text. We hope this addition makes the general anatomy of the larva more clear.

*Since the neurons are unbranched, do they run in bundles with a constant set of neighbors as they do in C. elegans?*

Although there are axon bundles containing neurons of specific types, this bundling is not strict and neurons braid within axon bundles, with many exiting their bundle and change neighbours along their lengths, unlike the case in *C. elegans*. We have added the following text in Results:

“Axons fasciculate in bundles but braid their positions within their bundle or sometimes defasiculate.”

*There should be a description of the larva's behavior. One finds out in the Discussion (fifth paragraph) that it does not feed! That's going to make a big difference in the complexity of the nervous system. What does it do besides swim? Does it chemotax, thermotax, and so forth?*

These are global questions to which we can respond only in brief in our text. We regret the lack of a comprehensive review to which we could refer. See also Reviewer 1 comment 6. We indicate that larvae do not feed in this comment (non-feeding lecithotrophic larvae). We add the following text in the Introduction: “In addition to phototactic and geotactic behavior, there is evidence of chemotactic behavior just before settlement (Svane and Young, 1989) and of some mechanosensory responses in swimming larvae (Bone, 1992).”

The main goal of swimming is thought to be that of dispersal, yet field studies and models also reveal that many animals have a dispersal distance of less than 6km/generation from adult colonies (Svane and Havenhand, 1993; Petersen and Svane, 1995; Kanary et al., 2011). Some eggs and hatched larvae also remain in a mucous string secreted by the adult (only 40-60% secreted in such a string actually escape the mucous, further limiting their dispersal.

In fact, active swimming was not considered in a recent dispersal model for ascidians despite their sometimes long swimming period (3-6 days) because their mean speeds are only 0.6-4mm/s at bursts of 5-20 seconds (Tsuda et al. 2001; Tsuda et al. 2003; McHenry and Patek 2004; Zega et al. 2006), which are negligible compared with the local current velocity of 27mm/s (Kanary et al., 2011). Thus, rather than retaining networks for active dispersal by swimming, retention of a complex network underlying larval behaviours may be selected for by survival pressure. We add the following text in the Introduction: “Because larvae do not feed, their main biological imperative is survival and successful settlement to undergo metamorphosis into a sessile adult, in an environment with appropriate food and reproductive resources. Thus, entering the water current and avoiding predation by filter feeders may be the foundation for the larva’s many behavioral networks, especially in early life before settlement.”

*What sensory-behavioral pathways are revealed in the connectome? What circuits governing swimming behavior and its control?*

We add the following text to the Discussion to address this comment: “Mechanosensory drive of symmetrical swimming was initially dismissed because tails can retain swimming independent of the head, and no known input from mechanosensory cells to motor neurons was known. However, we find multiple inputs from mechanosensory tail neurons’ interneurons to motor neurons, both from AMGs in the dorsal MG, and as reported from the bipolar tail neurons (Stolfi et al., 2015). It is not yet clear whether the cilia of DCENs or VCENs may alternatively have thermoreceptive, electroreceptive, or chemoreceptive properties, although there is evidence for chemotactic behavior during larval settlement, mediated by epidermal sensory neurons.”

We also added a summary figure to the following text to relate connectome pathways to specific sensory modalities.

“The shortest sensory pathway to any motor neuron connects via a brain vesicle interneuron, and is thus disynaptic, although most direct pathways involve two interneurons and are thus trisynaptic (Figure 11). However, these shortest paths fail to depict the complexity of integration revealed in the total network (Figure 7—figure supplement 1; Figure 9—figure supplement 1).”

“Figure 11. The shortest CNS pathways between sensory neurons and motor neurons for different sensory modalities are three-synapse arcs. […] These pathways are all interconnected, overlapping networks for all sensory modalities thus underlying the complex reality of the actual behavioral network (Figure 9—figure supplement 1).”

*Here are some additional questions or comments that occurred to this reviewer:*

*1) What constitutes the CNS and the PNS? CNS is referred to early in the paper without definition. Figure 15 shows that there are also a small number of PNS neurons. How are these distinguished in such a small nervous system? There is an error in the second paragraph of the Discussion, where it is stated there are 302 neurons in "most of the CNS" of C. elegans. There are 302 neurons in the entire nervous system and there is no clear division in C. elegans between CNS and PNS; these terms are not used.*

We acknowledge this distinction by indicating that the 302 neurons refers to most of the nervous system of *C. elegans* whereas 177 neurons refers explicitly to the CNS of *Ciona*. In addition, we have added the following text in the Results:

“These regions were additionally innervated by peripheral neurons with cell bodies residing in the epidermis that extend cilia into the tunic. Thus, unlike *C. elegans*, which lacks a distinction between a CNS and peripheral nervous system (PNS), *Ciona’s* chordate nervous system has clear divisions between the CNS and PNS. We annotated only those PNS neurons that innervate the CNS (Figure 1).”

*2) Neuron types: this concept is used throughout, but how neuron "type" is defined is not described. Do the neurons really fall into a set of discreet, unambiguous types? In C. elegans, the broad classifications sensory neuron, interneuron, motorneuron are not unambiguous. Many neurons generally considered to be interneurons because of the large number of synapses they make with other neurons also make nmj's, while many neurons considered to be motor neurons also have lots of synapses onto other neurons. "Motor neurons" can have stretch receptors, and "sensory neurons" can have synapses onto muscle. Careful examination reveals that where the cut-offs should in fact be among these broadest classes is unclear and may be somewhat arbitrary. When the manuscript first refers to different neuron types (subsection “Neurons”: brain vesicle intrinsic interneurons, MG interneurons, BV relay interneurons), the reader does not know what these neuron types are. So the descriptions of their various properties are hard to appreciate. Diagrams of the various types, what defines them, where their cell bodies lie, and where their processes go, are needed.*

We agree with the reviewer on this point. The primary criteria for each cell type are: a) the location of the soma along a rostrocaudal axis, whether it falls into the brain vesicle, neck, motor ganglion or caudal nerve cord; its termination in these same regions; and its connectivity. Some neurons have specialized structural features, as for example whether the neurons have cilia or sensory neurons that have specific structures. The revised key (Figure 1—figure supplement 2) helps to enumerate and clarify these features.

*3) In the first paragraph of the subsection “Neurons”, how can a "terminal" make an en passant synapse?*

See our response to reviewer 1, response 4, as follows:

“Many terminals had but small swellings, and their synapses occurred onto collateral terminals or axons, so were considered en passant.”

*4) In the third paragraph of the subsection “Synapses”; junctions <2 sections were omitted from consideration. What is the justification for this cutoff? Why not junctions <3? How many junctions does this exclude and what fraction of the "load" do they carry?*

The justification was of course mostly to ensure greater accuracy in our connectome, by requiring connections to be validated by >2 synaptic profiles between the same partner neurons. By removing single-profile synapses we eliminated 18% of all synaptic partnerships, and by removing all 2-profile synapses we would have eliminated a total of 35% of all synaptic partnerships. We now add the following text to the legend of Figure 3:

“Removing single-profile synapses eliminates 18% of all synaptic partnerships, and removing all 2-profile synapses would have eliminated a total of 35% of all synaptic partnerships.”

*5) In the fourth paragraph of the subsection “Synapses”, and throughout the paper: average is used as a statistic (average 37 presynaptic sites per CNS neuron). This doesn't seem to be a meaningful statistic. The distributions need to be given. In this regard, Figure 3 should probably be a semi-log survival curve. The curve shown doesn't tell us much as most of the points either don't move much along the X-axis or much along the Y axis.*

On reflection we agree with this point and now give the number of synapses per neuron as a mean and standard deviation, with the corresponding range, and have added/amended the following text:

“Each CNS neuron thus formed on average 49+61 (SD) presynaptic sites with a range of between 1 and 430 synapses and an average of 13+23 (SD) putative gap junctions, with a range between 0 and 166. Each postsynaptic neuron received an average of 39+42 (SD) synapses from each presynaptic partner, with a range between 0 and 179.”

*6) Figure 3 is described in the text (subsection “Synapses”, fifth paragraph) as showing the number of presynaptic sites versus the number of postsynaptic partners, but the axes in the figure are labeled number of synapse profiles vs number of synapses. What are synapse profiles?*

We agree, and have exchanged the original Figure 3 for a plot that shows depth of synaptic contacts, as explained in response 5 to reviewer 1. We now add a new Figure 3 which plots the numbers of synapses as a function of the numbers of synaptic partnerships. We now cite this Figure in the text in the place of the former Figure 3, and trivially revise the text to read:

“For each neuron, the number of presynaptic sites varies with the number of its postsynaptic partners, plotted for all neurons and their synapses, but excluding the neuromuscular junctions (Figure 3).”

*And why is this ratio of interest? What's the point being made here, that all neurons have the same density of synapses?*

This is related to the previous point. The ratio is of interest because it indicates that each of its partners increases the number of synapses made by a presynaptic neuron, and thus reflects a postsynaptic drive to the total synapse load. We now add this in the following short text:

“This relationship indicates that each of its partners increases the number of synapses made by a presynaptic neuron, and thus reflects a postsynaptic drive to the total synapse load.”

*7) In the second paragraph of the subsection “Sidedness in the brain vesicle”: there are said to be 17 enigmatic coronet cells on the left side, and Figure 5 is referred to. But I cannot see 17 enigmatic coronet cells in this figure. In fact, it's hard to distinguish any cells in the figures produced by Reconstruct (I presume). There need to be drawn diagrams to accompany them. And the drawings should be shown in the context of the entire larva, like the drawings of C. elegans neurons in WormAtlas (e.g. http://wormatlas.org/neurons/Individual%20Neurons/AVAframeset.html).*

This is a difficult point to address. We agree about Reconstruct and could wish that our study would have been commenced later, when superior reconstruction software would have been available to us. Figure 5 is an honest presentation of what a real reconstruction looks like without editing. The figures in Wormatlas are beautiful drawings, but that is an extensive document. We do now illustrate the coronet cells in our new Figure 1 and in Figure 7, which is also highly rendered, where their position is illustrated by orange profiles. To reveal the coronet cells in particular we therefore now render them from a dorsal view in a new panel, Figure 5, to address the reviewer’s point.

8) Figure 12. Can't read the labels in A and B. How do we know these left/right differences aren't just developmental noise?

Labels in A and B of what is now Figure 13, have been changed to white on blue, to enhance their chromatic contrast. We don’t in fact know that the left/right differences aren't just developmental noise, which would require the reconstruction of multiple larvae at different stages. We allow the possibility of developmental noise in the following addition (Discussion):

“Additionally, we cannot exclude the possibility that minor left/right asymmetries might be the product of developmental noise.”

*9)In the sixth paragraph of the Discussion: number of neuron types. This seems to me to be Results, not Discussion.*

It could be either Results or Discussion. These types are now provided in the key (Figure 1—figure supplement 2) referred to within the Results. We put it in the Discussion chiefly to compare with *C. elegans*.

*10) I can't relate the cells listed (or cell types?) in the legend to Figure 7 to the cell types (?) listed in Table 2. I can't relate the cells listed in the master cell list Figure 3—figure supplement 1 with the cells e.g. in Figure 3. The master cell list Figure 3—figure supplement 1 is said to be an excel file, but mine's a PDF. Excel file would be much easier to work with (e.g. the heading row can be kept visible as one scrolls down the list).*

This was our fault. The abbreviations now appear in the key (Figure 1—figure supplement 2), and we have changed the figures so that abbreviations and colours for the cell types are now cross-referred to the key. Additionally, we have now included small thumbnails of individual neuron reconstructions (whole cells, or terminals for photoreceptors) in Figure 3—figure supplement 1.

*11) I don't have a Figure 9—figure supplement 1.*

Likewise, this was our fault, the supplement was omitted from the submission, and is now Figure 9—figure supplement 1 (see response to comment 8 of reviewer 1).

*12) Table 1 lists 3 polyadic gap junctions, but polyads are not defined for gap junctions because there is no defined pre- and post-synaptic cell.*

We could not find a more suitable descriptive term for membrane appositions (GJs) that link more than two elements at a single site. These number 3% of the total number of such appositions. Given the uncertainty anyway in identifying GJs from membrane appositions we prefer not to remake the table for this point.

*13) Figure 15. This is the main presentation, along with individual neuron maps (missing), of the findings of this reconstruction. This is presumably the directed graph of chemical connections. The corresponding undirected graph of the gap junction connections needs to be included.*

We have now included small thumbnails of individual neuron reconstructions (whole cells, or terminals for photoreceptors) in Figure 3—figure supplement 1. We have also now included the corresponding undirected graph of putative GJs as Figure 16—figure supplement 2.

*I don't understand why the weights of the matrix are given as "percentile synaptic strength." In fact, I don't know what this means, rank order percentile? or percent of the highest value? Is a matrix with numbers (number of EM sections, or physical size in microns, since section thickness varied across this reconstruction) available rather than these colors? Modelers or theoreticians who would like to study the properties of the network (there will be many!) will need this.*

We have now remade both matrices of chemical and electrical synapses, using the color-coded depth by cumulative depth of synaptic contacts rather than percentile synaptic strengths. Colors represent ranges as illustrated in the new key.

*Reviewer #3:*

*This is a beautiful and very important work describing the entire neuronal connectome of the Ciona tadpole larva. Ciona early development and neuronal development are very well understood and the full larval connectome is a milestone addition to the resources available for this important chordate model organism. Importantly, urochordates are the sister group to the vertebrates, and have been studied intensively to understand the evolution of key vertebrate features. The connectome also provides a very important reference point for broader circuit comparisons across animals.*

*Given that this is a full connectome paper, I think the authors should not feel constrained by the 5000 word limit, but aim to describe the neuron complement and the full circuits in more detail. They should also think about how they can relate the connectivity to known larval behaviors (e.g., phototaxis, gravitaxis, larval settlement).*

We have now extended the length of the text, albeit conservatively, and are glad to have this reviewer’s and the editor’s sanction to do so.

*First, larval behaviors:*

There is one important mistake that the authors need to correct first to be able to better interpret the connectome.

*They state in the paper twice that the tadpole swims in a bilateral fashion with tail undulations: "The apparent symmetry of the characteristic swimming pattern of chordate-like tail undulations" and "Larvae exhibit a simple bilaterally symmetrical swimming response to light "*

*This is not true, Ciona and other ascidian larvae swim in a helical pattern. This was already known to Eakin and Kuda (1971) who wrote "When an ascidian larva is actively swimming the body rotates about its long axis." More importantly, a detailed kinematic analysis of swimming of the Ciona tadpole revealed how the larva turns towards light during helical swimming.*

*The authors should consult and cite "The kinematics of phototaxis in larvae of the ascidian Aplidium constellatum" by McHenry and Strother (Marine Biology, 2003) and "The morphology, behavior, and biomechanics of swimming in ascidian larvae" by McHenry (Can. J. Zool. 83: 62-74 2005). Appreciating the correct motor pattern is important to correctly interpret how the circuits work. The helical pattern emerges due to the asymmetry in tail bending. This is likely one of the main reasons for the high degree of asymmetry in this animal.*

This is an essential point. Our reference to bilateral swimming was to the pattern of muscular contraction as recorded, for example, in Nishino et al. (2010) their Figure 2, rather than to larval trajectory, as reported by Eakin and Kuda and in the papers of McHenry and colleagues. We should acknowledge that we were of course aware of the latters’ reports, but these were cut from the final version in view of the word limit. Now that we include more detail on larval behavior, it makes sense to relate our circuits to the helical pattern of larval swimming. We do this in the following text:

In the Introduction we write: “This sidedness may also correspond to larval behavior because ascidian larvae pursue a helical trajectory when swimming (McHenry and Strother, 2003; McHenry, 2005) and ascidian larvae are thought to use klinotaxis to respond to visual cues by modulating the symmetry of tail kinematics (McHenry and Strother, 2003).”

We continue in the Discussion with the following amended sentence: “Larvae swim in a helical pattern (McHenry, 2005) from simple bilateral flexions of the tail (Video 1: Bone, 1992) responding to light and a directional response to gravity (Tsuda et al., 2003).”

On the other hand, we do not entirely agree that the “Appreciating the correct motor pattern is important to correctly interpret how the circuits work” because asymmetry may be introduced by the tail muscle itself, which may even be more likely in muscle cells that are held in a curved posture, and may not reside entirely, or even at all, in the motor circuits that innervate the muscle. We don’t believe the last proviso, but think it sufficient simply to refer to the pattern of helical swimming, leaving any functional outcome hanging for the moment. On the other hand, we think helical movement bears importantly on the pattern of illumination it provides to the ocelli, when the shadow of the pigment crosses the outer segments. We cover these points in the following revised of the Discussion:

“Larvae swim in a helical pattern (McHenry, 2005) from simple bilateral flexions of the tail (Bone, 1992; Nishino et al., 2010). […] Based on gravitaxis in young larvae (Tsuda et al., 2003) we also predict that the antenna neurons’ circuits underlie a directional response to gravity.”

*Video 1 does not demonstrate bilateral swimming, it is actually impossible to tell from this video what the locomotor pattern is (bilateral or helical) because the larva moves out of focus and the frame rate is too low. This video may be removed or just used as an illustration.*

We agree and have substituted a high-speed video, which is used simply as an illustration for readers who may never have seen larval swimming. Reference to this video is now omitted from the Discussion.

*Neuron types and connectivity:*

*Figure 3—figure supplement 1 lists all cells and their important features. Since it is a comprehensive paper, the morphology and position of at least one example from each cell type should be shown in a supplementary figure (or figures) and where each cell type is located in the body (show body outline).*

*Likewise, some of the anatomical diagrams are difficult to interpret because the cells are not shown in the context of the whole body or together with other cells from the same circuit.*

*Same for the videos: Video 2 and Video 3 – the detail of the ocellus should be embedded into the larger anatomical context (e.g., by showing the body outline).*

We have addressed this concern also raised by reviewer 1 in our response to that reviewer: “An additional, very large version of this key with annotated cartoons of neuron shape will be deposited in Open Connectome” and by reviewer 2, in our response to that reviewer. “The body outline is also now shown (for the BV, neck and MG), along with a rendered version of cell bodies for Video 2 and Video 3)”.

*It is in general difficult to follow the description of connectivity of the different subsystems (e.g., eyes, otolith), because the relevant details are embedded in the summery diagrams and are not explained in full. I would really encourage the authors to describe the different sensory-motor systems in more detail.*

This comment relates to others from reviewers 1 and 2. We have summarized pathways pictorially in a new figure (Figure 11). We do not agree that text is more clear than figures, and think instead that this new figure goes a long way to making the existing text more readable, as indicated in response to Reviewer 2’s comment given above. Further text could we think only increase the length and opacity of our already long paper. The primary problem is simply that the full CNS connectome is very complex, but that each sensorimotor network can ultimately be reduced to a trisynaptic pathway, and this we have now tried to present more clearly. One of the chief issues in presenting a more complete network for *Ciona* is that, unlike the connectome for *C. elegans*, we still lack essentially all functional information on each type of interneuron and its particular neurotransmitter identity.

*This should include the reconstructed neurons in the whole-body context (anatomical diagrams) and the corresponding circuit diagrams. For example, what is the full cell complement and wiring diagram of the PRC circuit? Show type I, II, III PRCs and their downstream circuitry, to the muscles. Likewise, show the otolith and coronet circuits. What are the PNS neurons and their circuitry? How about the tail circuitry that probably tunes the left-right motor pattern?*

We agree with the reviewer, and Figure 11 now has the simplified summary of these pathways, and we do now also include a new video (4) that shows an animation of the visual pathway with cell bodies as spheroids within the outline of the CNS. We also report the antenna cell (otolith) pathways in Figure 10—figure supplement 2. But a more comprehensive treatment would require an addition equal in length to the existing text, and really lies beyond the scope of this first submission.

*The integration can then be described following the details of the subsystems (as is now e.g., in Figure 9).*

This, we think, is simply beyond the scope of a single paper, much as we would like to take an entire issue of *eLife* to accommodate the request.

*It would also be very interesting to see the strength of all reciprocal connections (e.g., represented as the geometric mean of 2-way synapse numbers) at the level of the entire connectome. This can easily be computed from the whole matrix.*

Could calculate

*It is hard to read Figure 7—figure supplement 1. The network should be shown in higher magnification. An alternative representation that would be useful would be to merge cells of the same type and connectivity into one node and show the average synaptic strength.*

We have now added a further network diagram (Figure 9—figure supplement 1) in which we pool neurons of the same class into a single node and summed their synapse numbers. This comment was not quite clear to us. Given that many cells of the same class converge upon a single postsynaptic target, it makes more sense to us to refer to the pooled synaptic strength than to show the average synaptic strength. See also our response to Reviewer 2.

*Figure 11. Also show cell morphology of MN classes. Please also show a schematic of the different muscle types and their position relative to the MN axons. It is not clear how an MN projecting on the left side can synapse on muscles both on the left and right sides of the body.*

We have now included a figure showing the pairs of motor neurons reconstructed (Figure 12—figure supplement 1) as requested. The general pattern of innervation for different motor neurons and the arrangement of the three different muscle bands is we think sufficiently reported in the literature (see for example: Nishino et al. 2010, their supplementary figures). We do not understand how our text could have been misinterpreted to indicate that a motor neuron projecting on the left side can synapse on muscles both on the left and right sides of the body, but must accept that it was unclear to the reviewer. We inserted the word “pairs” in the subsection “The motor ganglion”, in case this was the cause for the misunderstanding. All motor neurons project unilaterally, of course.

*Figure 15 is hard to read – please label neuron categories also on the presynaptic side. Provide list of abbreviations. Label major categories in the figure (corresponding to the boxes e.g., sensory neurons).*

We understand this problem. There are many cells and if all are to be presented in a single-page matrix, each necessarily will be small. We have added labels on the left-hand presynaptic side in a revised version of the figure and the abbreviations used are now provided in Figure 1—figure supplement 2, with colour coding of the cells that corresponds to the colours in the network diagrams. We have however retained the numerical labelling of the boxes in this figure, because to add text would have obscured the information in the matrix intercepts, which we needed to convey.

*Discussion:*

*The functional and comparative implications of the Ciona connectome should be spelled out in more detail. For example, how does the connectome explain negative phototactic behavior? The strategy of ascidian phototaxis is helical (not visual, for an explanation see Randel Philos Trans R Soc Lond B Biol Sci. 2016 doi: 10.1098/rstb.2015.0042.). It is known that upon light off, the tail begins to vibrate (Grave, 1921; Dilly, 1964). This would mean that when the helically swimming tadpole ocellus turns away from the light source, it initiates tail movement.*

We now cite the important paper by Randel and Jékely, the publication of which overlapped the period of preparation for our own manuscript. We cannot of course respond to the question of how does the connectome explain negative phototactic behavior, because our data are entirely anatomical and to seek such an explanation immediately leads us into undue speculation. We share the reviewer’s curiosity, of course, and even went so far as to generate the following text, which we have since removed, but which illustrates the uncertainties in all such speculations from anatomical connectome data:

“helical phototaxis could be explained by a simple circuit from hyperpolarizing glutamatergic photoreceptors onto possibly excitatory brain vesicle relay neurons, which in turn are presynaptic to cholinergic motor ganglion interneurons on the left and right sides, with anatomically greater synaptic drive on the left side than the right (ocellus) side that are themselves presynaptic to cholinergic motor neurons; thus allowing shading to differentially drive muscle contraction on both sides. What is unclear, however, is what role anatomical circuit asymmetries actually play in phototaxis as periodic shading activates and deactivates photoreceptors, given the differences between neurotransmitter phenotypes of different photoreceptors and the complexity of their downstream pathways.”

We are however encouraged by the thought that the identification of visual interneurons we now provide, can be combined with genetic intervention studies to generate appropriate functional studies.

*What can one deduce from the connectome? How does it compare to other phototactic systems (fish, lamprey, annelid)?*

The authors interpret the one sidedness of the eye as a reduction in one side, which is likely correct, and may have been possible as the larva lost visual phototaxis (which requires two eyes) and evolved helical phototaxis.

*Gravitaxis has also been studied by otolith ablations. Is the motor pattern known? Does the connectome explain how gravitaxis works?*

Our response to these important issues raised by the reviewer is essentially the same as we give above. The complement of cells and their pathways in the visual network of larval *Ciona* is most similar to the level of complexity approaching visual phototaxis as illustrated by stage (f) or (g) in Figure 2 of Randel and Jékely’s paper. In relation to this figure, the complexity of pathways found in *Ciona* suggests to us that the unilateral ascidian larval ocellus may have arisen by loss of a contralateral ocellus in ancestral forms. We have added this idea more explicitly to the Discussion as a new paragraph, as follows:

“Given that ascidian larvae swim in a helical pattern (McHenry, 2005) and have a single-sided ocellus, their phototactic behaviour is of a helical, not visual nature, as defined by Randel and Jékely (2016). […] Helical swimming of the ascidian larva provides a mechanism by which a preexisting bilateral visual phototaxis circuit could have been co-opted into a complex hybrid helical phototaxis circuit, still allowing mechanisms such as delay, sensory integration, and modulation to take place, and unlike the more direct helical phototaxis mechanisms in ciliated forms, such as protists and trochophore larvae (Randel and Jékely, 2016).”

[Editors' note: further revisions were requested prior to acceptance, as described below.]

[…]

*Reviewer #2:*

*This paper is improved after the first revision. However, it remains difficult for the general reader to follow. The difficulty lies in the fact that it is basically descriptive with many different neuron types referred to.*

*For the general reader, there needs to be a more careful and complete description of the work. First of all, the paper does not present an "entire synaptic connectome" (Abstract) or "full synaptic connectome" (first sentence of Discussion). It presents only a partial connectome, that of the CNS.*

We do appreciate that our text will not be an easy read for many readers, because the system is unfamiliar to most, even to most workers in the tunicate field. However, most readers would, we think, understand connectome as CNS connectome. We should also point out that our connectome provides most of the PNS connectome, because it identifies PNS neurons that provide input to CNS cells and that overlie the trunk and tail epithelium. To address the reviewer’s concern, however, we have nevertheless inserted ‘CNS’ as a qualifier to ‘connectome’ in the Impact Statement and in two later places, in the Abstract and in the thirteenth paragraph of the Discussion, where we refer to a ‘connectome’.

We also revised the composition of Table 2, adding two columns to provide segregated data for the Full network and for CNS neurons only.

Reflecting this reviewer’s concern, and our own interest, we have also now somewhat revised the title of our submission to become “The CNS connectome of a tadpole larva of *Ciona intestinalis* highlights sidedness in the brain of a chordate sibling”.

*This is just the beginning of how the presentation gets the reader off the track. Subsequently, CNS, PNS (also described as "epidermal"), and non-neuronal ependymal and support cells are all mixed in together in the descriptions.*

We now define and describe PNS earlier in the text.

The mixture of PNS, and designations as "epidermal", non-neuronal ependymal and support cells the reviewer finds is mostly a problem that cell types are not always segregated from each other in the larval CNS, as they would be in other nervous systems. For example, epidermal neurons are so identified based on previous accounts (Takamura, Imai and Meinertzhagen) that we chose not to revise. A subset of these epidermal PNS cannot be overlooked because they have axons that terminate and make synaptic connections in the CNS.

Ependymal cells are defined in the subsection “Synapses” (“those ciliated cells abutting the canal that lack an axon”).

The confusion between 300 and 177 is that not all cells are neurons, of course, as we remind the reader with the following change: “and constituting the remainder, ependymal cells (those ciliated cells abutting the canal that lack an axon) and three cells of the CNS that are ambiguous, having presynaptic sites, but lacking a neuronal form”. The discussion of ependymal cells is important for the question of asymmetry because the total cell numbers exhibit symmetry that is not upheld when examining neurons only.

*While connectivity, network properties and structure, and neuronal pathways are described, much more of the manuscript is devoted to individual neuron descriptions.*

The many newly identified cell types in this nervous system and many general readers’ unfamiliarity with the larval tunicate nervous necessitate this description. While the network may be valuable, its biological significance relies on what and how components are connected.

*Indeed, in the subsection “Sidedness in CNS pathways”, referring to some of their pathway diagrams, they state "However, these shortest paths fail to depict the complexity of integration revealed in the total network." The real emphasis can be seen in the subsection “Asymmetry in cellular composition”, where it out with the sentence "The overall cell complement is closely similar on the two sides (left: 125; right: 129; midline 46)." Total: 300, not 177 as we read in the Abstract. It is really a paper on the cellular structure of the Ciona larval nervous system, with emphasis on its left-right asymmetry, as properly presented in the title. Most of the Discussion is devoted to the developmental origin of the asymmetry.*

We have clarified the book-keeping on our cell numbers, which we confess may have been complex and confusing, and was written with concision in view of the *eLife* word limit. We now indicate the left/right and middle cell numbers on as “including neurons. ependymal and accessory cells”.

*For what they're worth, I offer the following further comments and suggestions for the authors' consideration.*

*1) I still don't understand the significance of the issue of left-right asymmetry. In the first paragraph of the Introduction, after the sentence that ends "and so provide a useful model to study many aspects of brain asymmetry," I need a next sentence that begins: "This issue is important because…" In the Abstract: "Chordate in body plan and development, the larva provides by contrast an outstanding example of brain asymmetry." By contrast to what?*

We added the following sentence and reference to Duboc et al. (2015), as follows: “This issue is important because brain laterality has been associated with increased fitness for animal life” We split the paragraph at this point to make this the concluding short sentence, before resuming discussion of *Ciona*.

We gave now removed “by contrast” from the Abstract, and we write later in the Introduction “In contrast to the situation in most chordates….”

*2) The sentences that begin the Results section belong in Methods. Similarly, the second paragraph of the subsection “Synapses”. Results need to begin: "The nervous system of the Ciona larva consists of.…" We need a general description of the larva and its nervous system so that when the EM series (and everything else) are described we know where we are. For example, for the general reader, "starting at the level of the otolith pigment.…" is meaningless as we don't know where that is. The major nervous system partitions given in the first paragraph ("posterior motor ganglion," "anterior brain vesicle," "motor ganglion," need to be shown in Figure 1.*

*3) What does it mean in the subsection “Synapses”, that "A total of 301 cells of the CNS were imaged,"? Wasn't every cell in an EM cross section "imaged"? Do you mean reconstructed or traced?*

We meant that all 301 cells were imaged and we go on in the text to say: “Cells omitted from our EM series, those rostral to the otolith and caudal to the bipolar tail neurons (Imai and Meinertzhagen, 2007b; Stolfi et al., 2015), are presumed to account for the remainder of the >331 cells reported by Nicol and Meinertzhagen (1991) and thus to number at least 30”.

*The issue of the ependymal cells should not be taken so lightly. Indeed, the authors feel the need to define these as "those ciliated cells abutting the canal that lack an axon," but they only do so in passing in the subsection “Asymmetry in cellular composition”. And apparently this work shows some cells previously thought to be ependymal are in fact neuronal. In the subsection “Synapses” it is explained that there are at least 30 CNS cells that were not studied. As we don't know whether these are neurons, ependymal, or other support cells, we don't really know that "the entire CNS included 177 neurons". Maybe it includes 207.*

We now introduce ependymal cells earlier in the fourth paragraph of the subsection “Synapses”, which we think clarifies the account. We do not think the identification of ependymal cells is in doubt at EM level, only that a previous account at LM level could not resolve axons.

For the question of cell numbers raised by the reviewer, it is correct that our analysis does not include the possible tail neurons that lay beyond the caudal extent of our analysed series. In a rostral direction there are only two axons, which are included in the totals given in Table 3, and which come from somata beyond our section series. They are reported separately in [Supplementary-material SD3-data] and belong to BVIN class of neurons.

We make the following revisions in our paper to take account of these further bookkeeping issues: Impact statement: “Serial-section EM analysis uncovers the CNS connectome of a *Ciona* larva, the second of any entire nervous system, and exposes left-right asymmetries in its synaptic circuits.” Additionally we removed the word “entire” when used to qualify reference to the connectome.

*4) Relay neurons. In such a highly cross-connected connectome as this can the authors really distinguish "relay" interneurons from other types? Figure 11 labels some neurons "interneurons" and some "relay" neurons. This distinction needs to be discussed.*

We think this is justifiable, and use relay to identify neurons that extend from one brain region to another, as they would be identified in, for example, the vertebrate brain. This is now clarified in the following sentence of the new paragraph added to the Introduction as follows:

“The relay neurons of the posterior brain vesicle extend axons through the neck to the motor ganglion, which overlies the anterior portion of the notochord, and contains neurons of the motor system.”.

*5) Explain and emphasize at the top that PNS neurons are often referred to as "epidermal."*

We included the following sentence to address this point, as follows:

“In addition to the CNS several sensory epidermal neurons (ENs) of the peripheral nervous system (PNS) populate the dorsal and ventral axis of the larva in a rostrocaudal sequence, with axons running beneath the epidermis (Imai and Meinertzhagen, 2007b).”.

*6) In the second paragraph of the Discussion: delete "most of".*

We deleted “most of”.

*7) The authors say they have determined the connectivity of 177 neurons. But their table of network statistics, Table 2, says there are 213 nodes in the chemical network and 193 in the gap junction network. Please explain this discrepancy.*

As explained above: “muscle was indeed included (4 nodes, dorsal left, dorsal right, medial left and medial right), as were PNS neurons and basal lamina, one orphan photoreceptor terminal profile, and two ambiguous cells which lack axons but have presynaptic sites. To cover these additional cells we added the following: “and two cells of the CNS that are ambiguous, having presynaptic sites, but lacking a neuronal form (Figure 1—figure supplement 2 and [Supplementary-material SD3-data])”. We made a new network with just the CNS neurons and analysis of this network is now added in Table 2 for both gap junctions and chemical synapses.”